# SSIMBaD: Sigma Scaling with SSIM-Guided Balanced Diffusion for AnimeFace Colorization

**Junpyo Seo**
Department of Computer Science
Seoul National University
jpseo99@snu.ac.kr

**Hanbin Koo**
Department of Computer Science
Seoul National University
nagnebin@snu.ac.kr

**Jieun Yook**
Department of Computer Science
Seoul National University
yookje@snu.ac.kr

**Byung-Ro Moon**[*]
Department of Computer Science
Seoul National University
moon@snu.ac.kr

## Abstract

We propose a novel diffusion-based framework for automatic colorization of Anime-style facial sketches, which preserves the structural fidelity of the input sketch while effectively transferring stylistic attributes from a reference image. Our approach builds upon recent continuous-time diffusion models, but departs from traditional methods that rely on predefined noise schedules, which often fail to maintain perceptual consistency across the generative trajectory. To address this, we introduce **SSIMBaD** (**S**igma **S**caling with **SSIM**-Guided **Ba**lanced **D**iffusion), a sigma-space transformation that ensures linear alignment of perceptual degradation, as measured by structural similarity. This perceptual scaling enforces uniform visual difficulty across timesteps, enabling more balanced and faithful reconstructions. On a large-scale Anime face dataset, SSIMBaD attains *state-of-the-art structural fidelity* and *strong perceptual quality*, with robust generalization to diverse styles and structural variations. Code and implementation details are available at [2].

## 1 Introduction

The rapid growth of content industries such as webtoons, animation, and virtual avatars has intensified the demand for automatic generation of high-quality Anime-style images. Among the various sub-tasks, colorizing sketch images remains a labor-intensive step in the content creation pipeline, as line art lacks shading and color information, requiring significant manual effort from artists. Automating this process not only enhances production efficiency but also ensures visual consistency across frames and styles [7, 10].

Early colorization models have been predominantly based on Generative Adversarial Networks (GANs). For instance, [7, 27, 37] leveraged conditional GANs guided by sparse color scribbles as user-provided inputs. However, these methods rely heavily on user hints and are sensitive to scribble placement and spatial correspondence. To alleviate this, Lee et al. [18] proposed reference-based colorization using a Spatially Corresponding Feature Transfer (SCFT) module that extracts semantic correspondences between the sketch and reference images. Yet, their approach struggles under large domain gaps or structural mismatches, a challenge that persists across reference-guided generation settings [19].

---

[*]Corresponding author.
[2]https://github.com/Giventicket/SSIMBaD-Sigma-Scaling-with-SSIM-Guided-Balanced-Diffusion-for-AnimeFace-Colorization

39th Conference on Neural Information Processing Systems (NeurIPS 2025).

Recently, diffusion models have emerged as a powerful class of generative models capable of producing high-fidelity images while avoiding common GAN pitfalls such as mode collapse and training instability [9, 13, 28, 30]. In particular, [3] was the first to apply Denoising Diffusion Probabilistic Models [13] to anime face colorization. By leveraging pixel-level supervision and multi-scale structural similarity, they reported improvements in PSNR, MS-SSIM [33], and FID [12] over GAN baselines. However, their discrete-cosine-based forward noise schedule rapidly degrades SSIM in early timesteps and then flattens later, yielding uneven perceptual difficulty across the trajectory. This non-uniform degradation complicates reverse-trajectory learning and hinders recovery of fine-grained color textures [31].

Alongside supervised diffusion, a line of *exemplar-driven, zero-shot* editing and transfer methods has gained traction in the Stable Diffusion (SD) / Latent Diffusion Model (LDM) ecosystem [25]. Cross-Image Attention [1], Attention Distillation [39], and StyleAligned [11] exploit attention maps (reflecting semantic correspondences across domains) and feature/statistics modulation (e.g., AdaIN) to perform appearance transfer without task-specific training. While appealing, these approaches are typically tied to large SD/LDM backbones, may require text prompts or additional conditioning, and incur considerable inference latency. In contrast, ControlNet [36] introduces trainable control branches on top of SD and does require fine-tuning but still inherits heavy foundation-model inference. In this work, our goal is orthogonal: we pursue a *lightweight and supervised* colorization pipeline targeted at near real-time inference and faithful recovery of high-frequency details. We nevertheless include these SD/LDM-based methods and a ControlNet variant in our evaluation for completeness and cross-paradigm comparison.

Elucidated Diffusion Models (EDM) [14] introduced a continuous-time noise parameterization in $\sigma$-space, enabling finer control over corruption and improved sample quality. Despite its strengths, EDM's standard (logarithmic) $\sigma$ sampling induces perceptual non-uniformity in colorization, where consistent difficulty across the trajectory is crucial.

To address this, we propose **SSIMBaD** (**S**igma **S**caling with **SSIM**-Guided **Ba**lanced **D**iffusion), a sigma-space transformation $\phi^*(\sigma)$ that aligns perceptual degradation *linearly* in SSIM, enforcing uniform visual difficulty across timesteps. Integrating this schedule into the EDM framework yields a continuous $\sigma$-space diffusion model tailored for Anime-style sketch colorization. The same SSIM-aligned schedule is used in training and in a lightweight reverse-only trajectory refinement stage: during training it prevents bias toward either near-clean or extreme-noise regimes; during inference it maintains consistent reconstruction difficulty over steps. Unlike prior methods that optimize reverse fidelity purely empirically, we explicitly anchor both forward and reverse dynamics to SSIM [33, 38]. On a large-scale Anime face dataset, this design achieves *state-of-the-art structural fidelity* while delivering *strong, competitive perceptual quality*, with robust generalization across diverse styles and structural variations.

Our main contributions are summarized as follows:

- **A Novel Unified Framework for Perceptually Balanced Diffusion** : We propose **SSIM-BaD** (**S**igma **S**caling with **SSIM**-Guided **Ba**lanced **D**iffusion, a pioneering framework that balances structural and stylistic fidelity in anime face colorization. Unlike prior approaches that suffer from inconsistencies in perceptual quality, SSIMBaD integrates perceptual schedule alignment, training-time consistency, and trajectory refinement to achieve stable and high-quality generation.

- **A Perceptual Sigma-Space Transformation for Enhanced Stability and Consistency** : We propose a novel sigma-space transformation, $\phi^*(\sigma)$, as a core innovation within SSIMBaD. By linearly aligning SSIM degradation across diffusion timesteps, this perceptual rescaling mechanism significantly improves step-wise consistency during the generation process, ensuring consistent perceptual generation, and overcoming the limitations of conventional noise schedules that often bias towards low or high-frequency details.

- **State-of-the-Art Performance Validated by Comprehensive Experimentation** : Extensive experiments on Danbooru AnimeFace show that SSIMBaD sets a new state of the art on PSNR, MS-SSIM, and FID, and delivers competitive LPIPS, CLIP-I, and DINOv2-I with strong cross-reference generalization. Ablations attribute the gains to all components: the EDM backbone, SSIM-aligned sigma scaling, and trajectory refinement.

## 2 Related Works

**GAN-Based Sketch Colorization**   Early colorization models primarily relied on GANs, guided by user-provided inputs such as sparse color scribbles [7, 27, 37]. While effective, these approaches are highly sensitive to scribble placement and often fail to generalize. To mitigate this, Lee et al. [18] proposed a reference-based method using SCFT module, which extracts semantic alignments between sketches and reference images. However, SCFT remains vulnerable to domain gaps and structural mismatches [19]. Other works explored semi-automatic pipelines [10] and two-stage GANs for flat-filling and shading [37], or incorporated text tags for semantic guidance [16], but challenges in consistency and stability persist.

**Generative Diffusion Models**   Diffusion models have emerged as powerful generative frameworks that address key limitations of GANs, including mode collapse and training instability [13, 29, 23, 9, 17, 26]. By learning to reverse a gradual noising process, they enable stable training and high-quality image synthesis. Nichol and Dhariwal [23] demonstrated that well-tuned diffusion models can outperform GANs across diverse benchmarks.

Subsequent advancements have improved their flexibility and performance. Song et al. [30, 31, 32] introduced a score-based formulation using stochastic differential equations (SDEs), enabling continuous-time generation and principled control over sampling dynamics. In parallel, several works have proposed deterministic sampling methods based on ordinary differential equations (ODEs), such as PNDM [20] and DPM-Solver [21], which accelerate inference while maintaining generation quality. Karras et al. [14] extended this with EDM, which operate in continuous $\sigma$-space and decouple noise level selection from timestep scheduling. EDM achieves state-of-the-art results on high-resolution datasets such as FFHQ [15] and ImageNet [8].

**Reference-Guided Diffusion Colorization**   Diffusion models have shown strong potential for image colorization by conditioning the denoising process on inputs such as sketches or reference images. Techniques like classifier guidance [9], cross-attention, and adaptive normalization enable fine-grained control. User-guided methods such as SDEdit [22] and DiffusArt [5] leverage partial noise or scribbles for controllable generation, but often require carefully crafted inputs. ILVR [6] and ControlNet [36] improve precision via reference alignment and auxiliary signals, yet depend on heavy Stable Diffusion / Latent Diffusion backbones [25].

Beyond these, SD/LDM-based zero-shot transfer methods—Cross-Image Attention [1], Attention Distillation [39], and StyleAligned [11]—propagate appearance via cross/shared attention and AdaIN, sometimes with text prompts or inversion, and are training-free but incur notable memory/latency costs. ControlNet, in turn, requires fine-tuning on SD while retaining foundation-model overhead. In contrast, we target a lightweight, supervised, jointly conditioned (sketch, reference) diffusion pipeline designed for near real-time inference and faithful high-frequency detail recovery; nevertheless, we include a modified ControlNet and the two zero-shot methods as baselines for a fair cross-paradigm comparison.

**AnimeDiffusion**   Cao et al. [3] propose a DDPM-based [13] pipeline for anime face colorization using a UNet denoiser with *early-fusion* conditioning: the sketch, the reference, and a noise-level map are channel-wise concatenated at the input rather than fused via cross-image attention or adapter modules. Training employs pixel-wise supervision and the forward corruption follows a discrete-cosine schedule. This raw-piling design is simple and efficient, relying on supervision to learn alignment without explicit correspondence layers.

## 3 Background: Elucidating the Design Space of Diffusion-Based Generative Models

The EDM framework [14] generalizes DDPM by introducing a continuous-time formulation of the forward noising process based on a scale variable $\sigma \in [\sigma_{\min}, \sigma_{\max}]$, which replaces the discrete timestep index $t$. Under this formulation, a clean image $x_0$ is perturbed into a noisy observation $x$ using a continuous noise level:

$$x(\sigma) = x_0 + \sigma \cdot \epsilon, \quad \epsilon \sim \mathcal{N}(0, \mathbf{I}). \tag{1}$$

This allows the model to learn over a continuous spectrum of corruption strengths, offering greater flexibility than DDPM's fixed timestep schedule. For notational simplicity, we denote $x(\sigma)$ as $x$ in the subsequent descriptions.

To stabilize training and ensure scale-invariant learning, the noisy input $x$ is preconditioned using the noise level $\sigma$ and a fixed constant $\sigma_{\text{data}}$ (typically 0.5). The network $F_\theta$ takes $x$ and $\sigma$ as input and produces a denoised estimate. The final prediction $D_\theta(x; \sigma)$, which serves as a function approximator of the clean target $x_0$, is computed using noise-dependent skip connections, as defined by:

$$D_\theta(x; \sigma) = c_{\text{skip}}(\sigma) \cdot x + c_{\text{out}}(\sigma) \cdot F_\theta(c_{\text{in}}(\sigma) \cdot x, \sigma), \tag{2}$$

where $c_{\text{skip}}$, $c_{\text{in}}$, and $c_{\text{out}}$ are predefined scaling coefficients derived from $\sigma$.

At inference time, EDM defines the generative process as a reverse-time probability flow ODE, derived from the SDE framework in score-based diffusion models [32]:

$$\frac{d\mathbf{x}}{dt} = -\frac{1}{\sigma}\left(D_\theta(x, \sigma) - x\right). \tag{3}$$

This ODE is numerically integrated using Euler or higher-order methods such as Heun or Runge-Kutta.

To discretize this continuous formulation, EDM introduces a $\rho$-parameterized noise schedule:

$$\sigma_i = \left(\sigma_{\max}^{1/\rho} + \frac{i}{N-1}(\sigma_{\min}^{1/\rho} - \sigma_{\max}^{1/\rho})\right)^\rho, \quad i = 0, \ldots, N-1. \tag{4}$$

By adjusting $\rho$, sampling steps can be concentrated in low- or high-noise regions. Most constants and scheduling heuristics in this formulation are directly adopted from the original EDM framework [14].

## 4 SSIMBad : Sigma Scaling with SSIM-Guided Balanced Diffusion for AnimeFace Colorization

We propose **SSIMBaD**, which incorporates a perceptually grounded noise schedule into the EDM [14]. Unlike prior log-based schemes, SSIMBaD aligns forward and reverse trajectories using a transformation that ensures perceptually uniform SSIM degradation.

The model conditions on $I_{\text{cond}} \in \mathbb{R}^{H \times W \times 4}$, formed by concatenating a reference image $I_{\text{ref}}$ and a sketch $I_{\text{sketch}}$. The clean target $I_{\text{gt}} \in \mathbb{R}^{H \times W \times 3}$ is corrupted with Gaussian noise to produce $I_{\text{noise}}$, which is denoised over time conditioned on $I_{\text{cond}}$. We now describe the key components of SSIMBaD, while Appendix A further elaborates on how sketches and reference images are transformed, as well as the data augmentation strategies applied.

### 4.1 SSIM-aligned Sigma-Space Scaling

The perceptual quality of diffusion models is highly sensitive to how noise is distributed across the denoising trajectory. In EDM, inference uses a $\rho$-parameterized schedule (4) to sample noise levels in a nonlinear manner, typically concentrating steps near low-noise regions. In contrast, training samples $\ln \sigma$ from a log-normal distribution $\mathcal{N}(P_{\text{mean}}, P_{\text{std}}^2)$, implicitly assuming a different transformation. This discrepancy implies that the transformation applied during training, $\phi_{\text{train}}(\sigma) = \log(\sigma)$, differs from that used in inference, $\phi_{\text{inference}}(\sigma) = \sigma^{1/\rho}$—resulting in a perceptual misalignment between forward and reverse trajectories.

To resolve this, we propose **SSIM-aligned sigma-space scaling**—a perceptually motivated strategy that defines a shared transformation $\phi : \mathbb{R}_+ \to \mathbb{R}$ used consistently across both training and inference. This transformation maps the noise scale $\sigma$ to a perceptual difficulty axis, ensuring visually uniform degradation throughout the diffusion process. Based on this transformation, we construct the noise schedule by interpolating linearly in the $\phi$-space:

$$\sigma_i^\phi = \phi^{-1}\left(\phi(\sigma_{\min}) + \frac{i}{N-1}\left(\phi(\sigma_{\max}) - \phi(\sigma_{\min})\right)\right), \quad i = 0, 1, \ldots, N-1. \tag{5}$$

To identify the optimal $\phi^*$, we consider a diverse candidate set $\Phi$ of analytic and squash-like transformations:

$$\Phi = \begin{cases} \sigma, & \log(\sigma), & \log(1+\sigma), & \sigma^2, & \dfrac{1}{\sigma}, & \dfrac{1}{\sigma^2}, & \mathrm{arcsinh}(\sigma), & \tanh(\sigma), \\[2mm] \mathrm{sigmoid}(\sigma), & \dfrac{\sigma}{\sigma+c}, & \dfrac{\sigma^p}{\sigma^p+1}, & \log(\sigma^2+1), & \arctan(\sigma) & & & \end{cases}$$

where $c > 0$ and $p > 0$ are tunable constants. Each $\phi$ is evaluated by how linearly its induced noise schedule aligns with perceptual degradation, measured by SSIM. Specifically, we compute the coefficient of determination ($R^2$) between $\sigma_i^\phi$ and SSIM degradation under additive noise:

$$\phi^* = \arg\max_{\phi \in \Phi} \mathbb{E}_{I_{\mathrm{gt}}, \boldsymbol{n}} \left[ R^2 \left( \left\{ \left( \sigma_i^\phi, \ \mathrm{SSIM}\left( I_{\mathrm{gt}} + \sigma_i^\phi \cdot \boldsymbol{n}, \ I_{\mathrm{gt}} \right) \right) \right\}_{i=0}^{N-1} \right) \right] \tag{6}$$

where $(I_{\mathrm{gt}}, \boldsymbol{n})$ are drawn from the data distribution and Gaussian noise, respectively.

We define the coefficient of determination as:

$$R^2 \left( \{(x_i, y_i)\}_{i=1}^{N} \right) \ = \ 1 - \frac{\sum_{i=1}^{N}(y_i - \hat{y}_i)^2}{\sum_{i=1}^{N}(y_i - \bar{y})^2}, \tag{7}$$

where $\hat{y}_i$ is the prediction of $y_i$ from the best linear fit to $\{x_i\}$ and $\bar{y}$ is their sample mean.

Our empirical search reveals that $\phi^*(\sigma) = \frac{\sigma}{\sigma+0.3}$ yields the highest $R^2$ and near-linear SSIM degradation. We adopt this transformation consistently in both training and inference, unifying the sampling dynamics across the diffusion process.

In addition, we replace the conventional $\log(\sigma)$ noise embedding with $c_{\mathrm{noise}} = \phi^*(\sigma)$ to align temporal conditioning with the perceptual trajectory. This alignment stabilizes training, improves reconstruction fidelity, and enhances generalization across diverse reference domains (see Section 5.3.1).

## 4.2 Framework of SSIMBaD

**Denoising Network** The denoising model $D_\theta$ follows a preconditioned residual design adapted from EDM [14], where the noisy input is scaled and fused with a learned residual correction. Distinctively, we replace the conventional $\log(\sigma)$ noise embedding with a perceptually grounded squash function $c_{\mathrm{noise}}(\sigma) = \phi^*(\sigma) = \frac{\sigma}{\sigma+0.3}$, ensuring better alignment with visual difficulty across the noise trajectory.

Formally, the denoiser is defined as:

$$D_\theta(I_{\mathrm{noise}}, I_{\mathrm{cond}}; \sigma) = c_{\mathrm{skip}}(\sigma) \cdot I_{\mathrm{noise}} + c_{\mathrm{out}}(\sigma) \cdot F_\theta \left( c_{\mathrm{in}}(\sigma) \cdot I_{\mathrm{noise}}, \ I_{\mathrm{cond}}; \ \phi^*(\sigma) \right)$$

.

**Training** To expose the model to a perceptually balanced distribution of noise scales, we sample $\sigma$ such that $\phi^*(\sigma)$ is uniformly distributed over $[\phi^*(\sigma_{\min}), \phi^*(\sigma_{\max})]$. The noise embedding $c_{\mathrm{noise}}$ is set to $\phi^*(\sigma)$, replacing traditional log-variance encodings. Given noisy input $x = I_{\mathrm{gt}} + \boldsymbol{n}$ with $\boldsymbol{n} \sim \mathcal{N}(0, \sigma^2 \mathbf{I})$, the pretraining loss is:

$$\mathcal{L}_{\mathrm{train}} = \mathbb{E}_{\phi^*(\sigma) \sim \mathcal{U}[\phi^*(\sigma_{\min}), \phi^*(\sigma_{\max})]} \mathbb{E}_{I_{\mathrm{gt}} \sim p_{\mathrm{data}}} \mathbb{E}_{\boldsymbol{n} \sim \mathcal{N}(0, \sigma^2 \mathbf{I})} \left\| D_\theta(I_{\mathrm{gt}} + \boldsymbol{n}, I_{\mathrm{cond}}; \sigma) - I_{\mathrm{gt}} \right\|^2. \tag{8}$$

**Trajectory Refinement** To further enhance perceptual fidelity, we apply trajectory refinement. The reverse diffusion process is initialized from a pure Gaussian noise sample $I^{(N-1)} \sim \mathcal{N}(0, \mathbf{I})$, and integrated backward using a perceptually scaled sigma schedule $\{\sigma_i\}_{i=0}^{N-1}$ derived from $\phi^*(\sigma)$. For each denoising step $i = N-1, \ldots, 1$ ($\sigma_0 = 0$), we perform **Euler** updates as:

$$I^{(i-1)} = I^{(i)} - \frac{\Delta t_i}{\sigma_i} \left( D_\theta(I^{(i)}, I_{\mathrm{cond}}; \sigma_i) - I^{(i)} \right), \quad \Delta t_i = \sigma_i - \sigma_{i-1}. \tag{9}$$

Instead of treating the entire process as fixed, we fine-tune the denoising trajectory itself by aligning the final reconstruction $I^{(0)}$ with the clean target $I_{\mathrm{gt}}$ via an $\ell_2$ loss:

$$\mathcal{L}_{\mathrm{trajectory \ refinement}} = \mathbb{E}_{I_{\mathrm{gt}} \sim p_{\mathrm{data}}} \mathbb{E}_{\boldsymbol{n} \sim \mathcal{N}(0, \mathbf{I})} \left\| I^{(0)} - I_{\mathrm{gt}} \right\|^2. \tag{10}$$

Here, $I^{(0)}$ denotes the reconstructed image obtained after performing the full reverse trajectory from $i = N - 1$ down to 0. This refinement can be interpreted as a fine-tuning step that encourages the entire denoising trajectory to terminate closer to the ground-truth image.

**Inference**   During inference, we reuse the same $\phi^*(\sigma)$ transformation and construct a deterministic schedule:

$$\sigma_i = (\phi^*)^{-1} \left[ \phi^*(\sigma_{\min}) + \frac{i}{N-1} \cdot (\phi^*(\sigma_{\max}) - \phi^*(\sigma_{\min})) \right], \quad i = 0, \ldots, N-1. \tag{11}$$

We then apply the same Heun's methodfollowing the formulation in EDM, as in trajectory refinement to produce the final image from pure noise.

## 5   Experiments

### 5.1   Dataset Description

We evaluate our method on a benchmark dataset introduced by [3], specifically curated for reference-guided anime face colorization. The dataset comprises 31,696 sketch–color training pairs and 579 test samples, all resized to a resolution of $256 \times 256$ pixels. Each training instance consists of a ground-truth color image $I_{\text{gt}}$ and its corresponding sketch $I_{\text{sketch}}$, generated via an edge detection operator such as XDoG [35]. The sketch images serve as the structural input, while the reference images provide appearance cues such as color and style.

We evaluate model robustness under two test settings. In the **same-reference** scenario, the reference image is a perturbed version of the ground-truth, sharing the same structural input as $I_{\text{sketch}}$. In the **cross-reference** scenario, the reference is randomly sampled from other test images, introducing variations in both color and facial attributes. This dual setup enables evaluation of reconstruction fidelity under ideal conditions and generalization under domain shift.

### 5.2   Evaluation Metrics

**Fidelity metrics** focus on low-level accuracy and structural consistency. PSNR measures pixel-level reconstruction quality via mean squared error, though it correlates weakly with human perception [2]. MS-SSIM extends SSIM across multiple scales of luminance, contrast, and structure [34], making it appropriate for sketch-conditioned colorization. FID computes the Fréchet distance between generated and real image features [12], capturing distributional realism and overall fidelity.

**Perceptual metrics** capture semantic and stylistic alignment beyond pixel fidelity. LPIPS quantifies perceptual distance using deep features [38], reflecting human judgments of texture and style plausibility. CLIP-I measures semantic consistency through cosine similarity of CLIP embeddings between generated and reference images [24]. DINOv2-I evaluates structural and style-level similarity using self-supervised visual features [4], offering a stable perceptual indicator less text-biased than CLIP.

### 5.3   Experimental Results

#### 5.3.1   Empirical Evaluation of SSIM-Aligned Sigma-Space Scaling Functions

Table 1: Transformation functions $\phi(\sigma)$ sorted by $R^2$ linearity with SSIM degradation. Bounded squash functions yield the highest perceptual alignment.

| $\phi(\sigma)$ | $\sigma^2$ | $\frac{1}{\sigma^2}$ | $\sigma$ | $\frac{1}{\sigma}$ | $\log(\sigma^2 + 1)$ | $\log 1p(\sigma)$ | $\text{arcsinh}(\sigma)$ | $\frac{\sigma^p}{\sigma^p + 1}$ |
|---|---|---|---|---|---|---|---|---|
| $R^2$ | 0.0616 | 0.0624 | 0.0768 | 0.1183 | 0.2225 | 0.3754 | 0.4001 | 0.7332 |

| $\phi(\sigma)$ | $\text{sigmoid}(\sigma)$ | $\frac{\sigma}{\sigma + 0.9}$ | $\tanh(\sigma)$ | $\frac{\sigma}{\sigma + 0.7}$ | $\log(\sigma)$ | $\frac{\sigma}{\sigma + 0.1}$ | $\frac{\sigma}{\sigma + 0.5}$ | $\frac{\sigma}{\sigma + 0.3}$ |
|---|---|---|---|---|---|---|---|---|
| $R^2$ | 0.6837 | 0.8196 | 0.8650 | 0.8710 | 0.8972 | 0.9275 | 0.9277 | **0.9793** |

To ensure perceptual consistency across the generative trajectory, we construct the noise schedule by uniformly sampling in a transformed $\phi(\sigma)$ space and applying its inverse. We empirically select

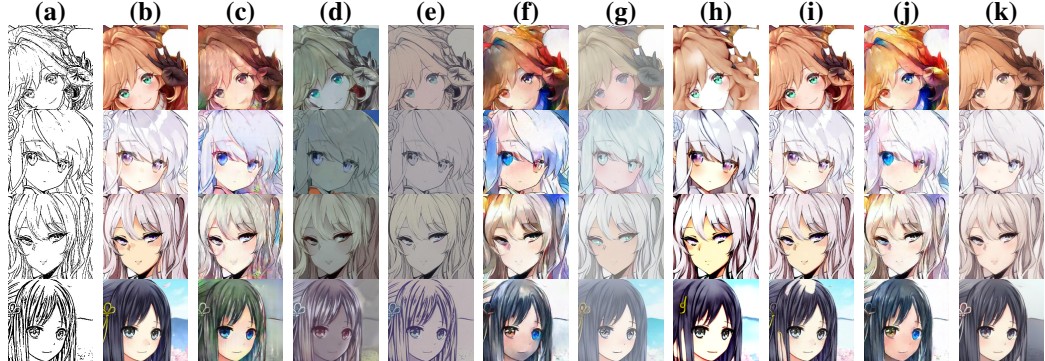

Figure 1: **Qualitative comparison of colorization results under the same-reference scenario.** From left to right: (a) Sketch input. (b) Reference image. (c) SCFT [18]. (d) AnimeDiffusion [3] (pretrained). (e) AnimeDiffusion [3] (finetuned). (f) AnimeDiffusion (EDM backbone, default $\sigma$-schedule). (g) ControlNet [36] (h) Cross-Image Attention [1]. (i) Attention Distillation [39]. (j) Our model (w/o trajectory refinement). (k) Our model (w/ trajectory refinement).

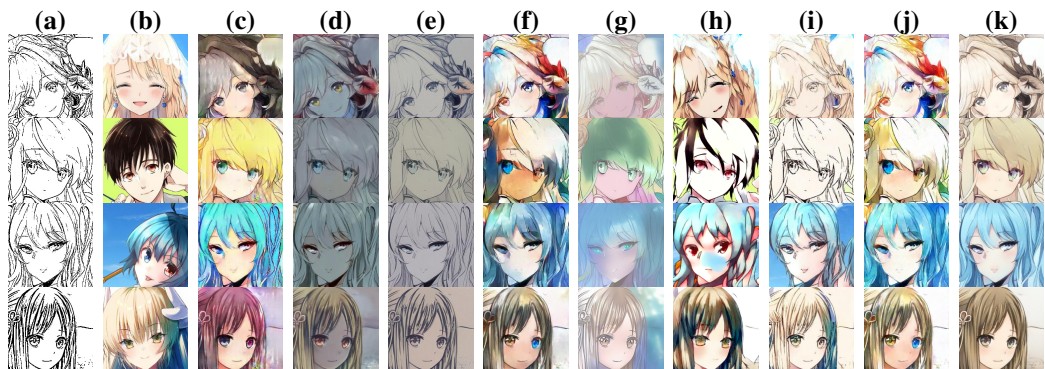

Figure 2: **Qualitative comparison of colorization results under the cross-reference scenario.** From left to right: (a) Sketch input. (b) Reference image. (c) SCFT [18]. (d) AnimeDiffusion [3] (pretrained). (e) AnimeDiffusion [3] (finetuned). (f) AnimeDiffusion (EDM backbone, default $\sigma$-schedule). (g) ControlNet [36] (h) Cross-Image Attention [1]. (i) Attention Distillation [39]. (j) Our model (w/o trajectory refinement). (k) Our model (w/ trajectory refinement).

$\phi(\sigma) = \frac{\sigma}{\sigma+0.3}$ based on its near-linear SSIM degradation behavior. Full analysis details are provided in Appendix B.

To construct a perceptually uniform noise schedule, we empirically analyze the relationship between SSIM degradation and transformed noise levels $\phi(\sigma)$ for various candidate functions. For each transformation $\phi$, a clean image $I_{\text{clean}}$ is corrupted at $N = 50$ different noise levels by adding scaled Gaussian noise as defined in (1).

### 5.3.2 Evaluation under Same and Cross Reference Scenarios

Table 2, together with Figures 1 and 2, supports a category-wise analysis of the evaluation. For clarity, baselines are grouped into four families: GAN-based (SCFT [18]), train-free attention (Cross-Image Attention [1], Attention Distillation [39]), light finetuning (ControlNet [36]), and the AnimeDiffusion family (pretrained, finetuned, EDM backbone [3, 14]). Our method (SSIMBaD) is reported with and without trajectory refinement. This organization makes explicit how architectural biases manifest as distinct trade-offs among structure, realism, and style, and it highlights why SSIMBaD achieves the most favorable overall balance.

The GAN-based model provides a stable classical anchor. SCFT [18] is competitive under the same-reference condition and remains relatively robust under cross-reference, yet it consistently

trails SSIMBaD on cross-reference fidelity. Style indicators are moderate. This pattern suggests that stability alone is insufficient to dominate when the reference is mismatched.

Table 2: Quantitative comparison under both same-reference and cross-reference settings across fidelity (PSNR, MS-SSIM, FID) and style-aware (LPIPS, CLIP-I, DINOv2-I) metrics. Best results per column are in **bold**, second-best are underlined.

| Method | Training | PSNR (↑) | | MS-SSIM (↑) | | FID (↓) | | LPIPS (↓) | | CLIP-I (↑) | | DINOv2-I (↑) | |
|---|---|---|---|---|---|---|---|---|---|---|---|---|---|
| | | Same | Cross | Same | Cross | Same | Cross | Same | Cross | Same | Cross | Same | Cross |
| SCFT [18] | 300 epochs | 17.17 | 15.47 | 0.7833 | 0.7627 | 43.98 | 45.18 | 0.1728 | 0.5008 | 0.9020 | 0.8247 | 0.9392 | 0.8622 |
| AnimeDiffusion [3] (pretrained) | 300 epochs | 11.39 | 11.39 | 0.6748 | 0.6721 | 46.96 | 46.72 | 0.2226 | 0.5107 | 0.8993 | 0.8203 | 0.9392 | 0.8576 |
| AnimeDiffusion [3] (finetuned) | 300+10 epochs | 13.32 | 12.52 | 0.7001 | 0.5683 | 135.12 | 139.13 | 0.2242 | 0.5069 | 0.8797 | 0.8012 | 0.9359 | 0.8554 |
| ControlNet [36] | 10 epochs | 14.74 | 12.08 | 0.7336 | 0.2007 | 40.20 | 50.25 | 0.2043 | 0.4930 | 0.9194 | 0.8311 | 0.9640 | 0.8739 |
| Cross-Image Attention [1] | free | 13.95 | 10.60 | 0.7147 | 0.4932 | 53.63 | 60.54 | 0.2661 | **0.4569** | 0.9369 | 0.8554 | 0.9335 | 0.8793 |
| Attention Distillation [39] | free | **19.58** | 10.08 | **0.8812** | 0.1252 | **32.93** | 94.17 | **0.1139** | 0.5385 | **0.9610** | **0.8819** | **0.9816** | **0.8941** |
| SSIMBaD (w/o trajectory refinement) | 300 epochs | 15.15 | 13.04 | 0.7115 | 0.6736 | 53.33 | 55.18 | 0.1878 | 0.4889 | 0.8975 | 0.8332 | 0.9339 | 0.8605 |
| SSIMBaD (w/ trajectory refinement) | 300+10 epochs | 18.92 | **15.84** | 0.8512 | **0.8207** | 34.98 | **37.10** | 0.1174 | 0.4804 | 0.9334 | 0.8508 | 0.9644 | 0.8826 |

Train-free attention methods emphasize semantic transfer via attention mechanisms without task-specific training on our data. As expected, this inductive bias aligns well with style metrics when the reference is aligned, but it does not reliably preserve structure or realism under mismatch. Attention Distillation [39] attains best-in-class same-reference columns for both structure and style, but degrades sharply under cross-reference (for example, MS-SSIM and FID deteriorate markedly), indicating brittle behavior when appearance cues no longer coincide with content. Cross-Image Attention [36] achieves the lowest cross-reference LPIPS, confirming its semantic alignment bias, but is less competitive on the remaining style indicators and on fidelity.

Light finetuning with ControlNet [36] strengthens several same-reference columns yet exhibits a clear drop under cross-reference. Style metrics remain decent, which suggests that short finetuning can amplify appearance cues while risking structural drift and realism loss outside the finetuned regime.

Within the AnimeDiffusion family, the pretrained model [3] offers a reasonable baseline. Simple finetuning raises PSNR and, in the same-reference case, MS-SSIM, but severely harms realism as reflected by large FID values, a characteristic fidelity-versus-realism failure mode. Replacing the discrete backbone with an EDM formulation [14] alone slightly perturbs perceptual alignment in the same-reference regime, which is consistent with the additional optimization burden of a continuous-time parameterization. Introducing SSIM-aligned sigma-space scaling on the EDM backbone subsequently recovers and improves structural fidelity, indicating that perceptually paced scheduling is a principal driver of reconstruction quality.

SSIMBaD combines an EDM backbone [14], SSIM-aligned sigma-space scaling, and a reverse-only trajectory refinement to deliver the strongest cross-reference fidelity (PSNR, MS-SSIM, FID) while remaining second-best in the same-reference setting; on style and perception it stays near the top (LPIPS [38], DINOv2-I [4]) with competitive CLIP-I [24], and qualitative results preserve geometry and transfer color without oversaturation. The mechanism is that SSIM-aligned scaling spreads perceptual difficulty evenly across timesteps, yielding steadier training signals and a smoother reverse trajectory that a lightweight reverse-only refinement can exploit without overfitting the forward corruption. This accounts for SSIMBaD's cross-reference advantage and consistently high style ranks, and it clarifies the robustness gap with train-free attention methods whose semantic focus lacks explicit structural pacing. In sum, across method families, SSIMBaD offers a balanced and robust solution that is better suited to practical scenarios with reference mismatch than approaches that peak on style but falter on structure or realism.

### 5.3.3 Comparison of Diffusion Schedules in DDPM, EDM, and EDM with SSIM-Aligned Sigma-Space Scaling

Figure 3 illustrates the behavior of the forward diffusion process for a single training image under different noise schedules. Specifically, it plots how SSIM values change across timesteps ($N = 25$) and visualizes a series of 50 corrupted images corresponding to each timestep, allowing intuitive assessment of the degree of corruption. These findings emphasize the crucial role of scheduling in aligning diffusion dynamics with perceptual difficulty.

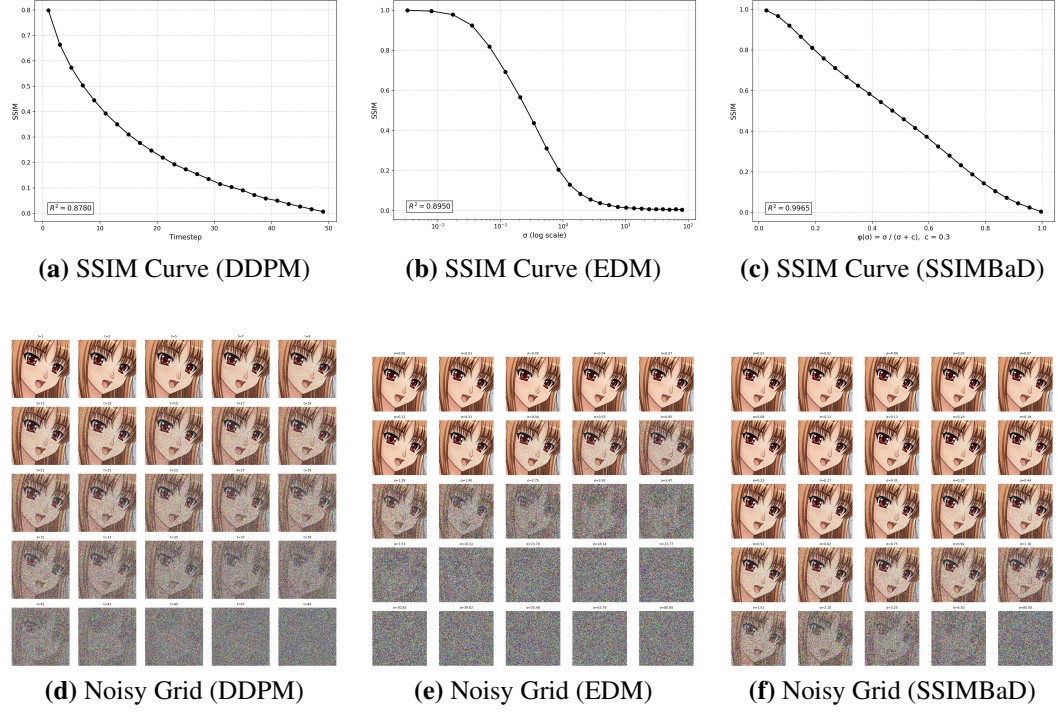

**(a)** SSIM Curve (DDPM)   **(b)** SSIM Curve (EDM)   **(c)** SSIM Curve (SSIMBaD)

**(d)** Noisy Grid (DDPM)   **(e)** Noisy Grid (EDM)   **(f)** Noisy Grid (SSIMBaD)

Figure 3: **Comparison of forward diffusion schedules.** Top: SSIM curves for DDPM **(a)**, EDM **(b)**, and our schedule $\phi^*(\sigma)$ **(c)**. Bottom: $5 \times 5$ corrupted grids **(d)**–**(f)** show each schedule's visual effect. Our method yields perceptually uniform degradation across timesteps.

The DDPM baseline employs a cosine-based schedule, designed to increase noise linearly across discrete timesteps. As seen in the graph in Figure 3-(a), DDPM introduces minimal noise during early steps but abruptly escalates noise levels in later stages, resulting in uneven SSIM degradation(noise levels) across timesteps. This leads to difficulty in reconstruction during the reverse process.

EDM improves upon DDPM by interpolating noise levels in $\sigma$-space via a $\rho$-parameterized schedule, yielding a smoother degradation curve (Figure 3-(b)). However, SSIM changes are concentrated in the mid-$\sigma$ range, with saturation at both ends. As a result, only a portion of the schedule contributes effectively to training, reducing overall efficiency and biasing learning toward the central region.

As shown in Figure 3-(c), the proposed $\phi^*(\sigma)$ schedule, which employs SSIM-aligned sigma-space scaling, is designed so that SSIM degradation becomes linear with respect to the transformation of $\sigma$. The images corresponding to each timestep demonstrate that, at no stage, is there an excessive SSIM degradation; rather, smooth and balanced noise is introduced at every step. This uniformity ensures that all diffusion stages become equally important, thereby improving reconstruction reconstruction fidelity across all frequencies. Furthermore, it enables more stable training and interpretable sampling behavior.

## 5.4 Ablation Study

Table 3 summarizes a cumulative ablation from the AnimeDiffusion baseline to three successive additions: an EDM backbone, SSIM-aligned scaling, and a lightweight reverse-only trajectory refinement. Moving to EDM raises PSNR in both regimes (same 11.39 to 13.30; cross 11.39 to 12.11) but slightly lowers MS-SSIM and worsens FID, consistent with the higher optimization burden of a continuous-time formulation paired with a schedule that is not perceptually paced. Adding SSIM-aligned scaling rebalances per-step corruption by difficulty and recovers structure (PSNR to 15.15/13.04; MS-SSIM to 0.7115/0.6736), while LPIPS decreases in both settings; however, FID has not yet improved over the baseline. The reverse-only refinement then converts this headroom

into the largest overall gains, yielding the best results across metrics (PSNR 18.92/15.84, MS-SSIM 0.8512/0.8207, FID 34.98/37.10) and pushing style scores near the top (LPIPS 0.1174/0.4804; DINOv2-I 0.9644/0.8826; competitive CLIP-I). Notably, FID improves by about 12 points in the same-reference case and 10 points in the cross-reference case relative to the baseline, and the gaps between same and cross shrink for MS-SSIM and DINOv2-I, indicating stronger style-preserving generalization. Overall, the pattern is consistent with a synergistic mechanism: SSIM-aligned scaling equalizes training signal across steps and smooths the reverse trajectory, and a small refinement confined to that trajectory delivers simultaneous gains in structure and realism, particularly under reference mismatch.

Table 3: Cumulative ablation study under both same- and cross-reference settings across fidelity (PSNR, MS-SSIM, FID) and style-aware (LPIPS, CLIP-I, DINOv2-I) metrics. Each added component incrementally improves model performance across all metrics and settings.

| Base | + EDM | SSIM-aligned sigma-space scaling | + Trajectory Refinement | PSNR ($\uparrow$) | | MS-SSIM ($\uparrow$) | | FID ($\downarrow$) | | LPIPS ($\downarrow$) | | CLIP-I ($\uparrow$) | | DINOv2-I ($\uparrow$) | |
|---|---|---|---|---|---|---|---|---|---|---|---|---|---|---|---|
| | | | | Same | Cross | Same | Cross | Same | Cross | Same | Cross | Same | Cross | Same | Cross |
| ✓ | – | – | – | 11.39 | 11.39 | 0.6748 | 0.6721 | 46.96 | 46.72 | 0.2226 | 0.5107 | 0.8993 | 0.8203 | 0.9392 | 0.8576 |
| ✓ | ✓ | – | – | 13.30 | 12.11 | 0.6426 | 0.6219 | 52.18 | 53.60 | 0.2192 | 0.4925 | 0.8886 | 0.8300 | 0.9217 | 0.8527 |
| ✓ | ✓ | ✓ | – | 15.15 | 13.04 | 0.7115 | 0.6736 | 53.33 | 55.18 | 0.1878 | 0.4889 | 0.8975 | 0.8332 | 0.9339 | 0.8605 |
| ✓ | ✓ | ✓ | ✓ | **18.92** | **15.84** | **0.8512** | **0.8207** | **34.98** | **37.10** | **0.1174** | **0.4804** | **0.9334** | **0.8508** | **0.9644** | **0.8826** |

# 6  Conclusion

This paper presented SSIMBaD, a diffusion framework for anime face colorization that reconciles training and inference with perceptual difficulty. The central contribution is SSIM-aligned sigma-space scaling, which reparameterizes the noise schedule to follow an approximately linear SSIM degradation, yielding uniform perceptual difficulty across steps. Coupled with an EDM backbone and a lightweight reverse-only trajectory refinement, the framework aligns forward corruption and reverse reconstruction along a single perceptual trajectory.

Comprehensive experiments on the Danbooru AnimeFace dataset validate the approach. Under same- and cross-reference conditions, SSIMBaD attains PSNR = 18.92/15.84 dB, MS-SSIM = 0.8512/0.8207, and FID = 34.98/37.10, outperforming SCFT, AnimeDiffusion, and a modified ControlNet baseline (Table 2). Against recent training-free methods, SSIMBaD maintains substantially stronger structure and realism under reference mismatch, e.g., cross-reference MS-SSIM 0.8207 and FID 37.10 versus 0.4932/60.54 for Cross-Image Attention and 0.1252/94.17 for Attention Distillation. Qualitative comparisons in Figures 1 and 2 corroborate these findings.

Ablation studies (Table 3) show complementary contributions. Moving to EDM improves fidelity; SSIM-aligned scaling equalizes per-step difficulty and stabilizes optimization; trajectory refinement then exploits the smoother reverse trajectory to improve realism and semantic coherence. Schedule analysis (Figure 3) confirms that the proposed scaling produces near-linear SSIM decay (high $R^2$), avoiding early under- and late over-corruption and improving step efficiency. Additional sensitivity analyses indicate robustness to solver choice (Euler, Heun) and step allocation, and faster convergence under equal compute.

Limitations remain in fine-grained details (e.g., small accessories, iris highlights) under extreme sketch sparsity or large reference–content gaps. Nevertheless, the empirical evidence indicates that aligning diffusion schedules with perceptual degradation is an effective and general principle. Beyond anime colorization, SSIM-aligned scaling is readily applicable to other conditional generation tasks that require structural preservation and perceptual balance, including sketch-to-image synthesis and controllable diffusion.

## Acknowledgments

The *ICT at Seoul National University* provides research facilities for this study.

This research was supported by the *Challengeable Future Defense Technology Research and Development Program* through the *Agency for Defense Development (ADD)*, funded by the *Defense Acquisition Program Administration (DAPA)* in 2025 (No. 915110201).

This work was also supported by the *BK21 FOUR Intelligence Computing Program* (Department of Computer Science and Engineering, Seoul National University), funded by the *National Research Foundation of Korea (NRF)* (No. 4199990214639).

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

# Appendix

## A   Details of the Proposed Framework

### A.1   Conditional Input Construction

Let $I_{\text{gt}} \in \mathbb{R}^{H \times W \times 3}$ denote the RGB ground-truth anime image, where $H$ and $W$ are the spatial resolution of the image. To form a pair of conditioning signals that guide both structure and style reconstruction, we derive two distinct inputs from $I_{\text{gt}}$: a structural sketch and a perturbed reference.

**Sketch Extraction**   The sketch $I_{\text{sketch}}$ is extracted via the extended Difference-of-Gaussians (XDoG) operator [35], which enhances edge-like regions through nonlinear contrast enhancement. Formally:

$$I_{\text{sketch}} = \text{XDoG}(I_{\text{gt}}) \in \mathbb{R}^{H \times W \times 1}. \tag{12}$$

This 1-channel sketch preserves high-frequency structure such as contours and character outlines, serving as a strong spatial constraint during generation.

**Reference Transformation**   To simulate reference-guided generation under diverse style domains, we construct a distorted version of $I_{\text{gt}}$ using a sequence of geometric transformations. First, a Thin Plate Spline (TPS) deformation is applied to introduce local warping, followed by random global rotations to inject non-aligned style cues:

$$I_{\text{ref}} = \text{Rotate}(\text{TPS}(I_{\text{gt}})) \in \mathbb{R}^{H \times W \times 3}. \tag{13}$$

This 3-channel reference encodes the target color palette and texture, potentially with mild spatial misalignments.

**Channel-Wise Conditioning**   The final conditional input is formed by concatenating the sketch and reference along the channel dimension:

$$I_{\text{cond}} = [I_{\text{ref}} \,\|\, I_{\text{sketch}}] \in \mathbb{R}^{H \times W \times 4}, \tag{14}$$

where $\|$ denotes channel-wise concatenation. This composite input retains both semantic layout and color style information, enabling the network to model structural consistency and stylization jointly. Note that $I_{\text{cond}}$ is held fixed throughout each diffusion trajectory to serve as a conditioning context for the denoiser.

### A.2   Incorporating EDM

We reformulate [3] within the continuous-time framework of EDM [14], preserving its U-Net-based conditional denoiser $F_\theta$ while adopting a noise-level parameterization based on a continuous scale $\sigma$ rather than a discrete timestep $t$. This transition from discrete to continuous noise coordinates enables finer-grained modeling of the forward and reverse processes, as well as improved control over perceptual degradation across the diffusion trajectory.

Under the EDM formulation, the forward process perturbs a ground-truth image $I_{\text{gt}}$ into a noisy observation $I_{\text{noise}}$ by adding Gaussian noise of standard deviation $\sigma$:

$$p_\sigma(I_{\text{noise}} \mid I_{\text{cond}}) = \int_{\mathbb{R}^{H \times W \times 3}} \mathcal{N}\left(I_{\text{noise}};\ I_{\text{gt}},\ \sigma^2 \mathbf{I}\right) p_{\text{data}}(I_{\text{gt}} \mid I_{\text{cond}}) \, dI_{\text{gt}}, \tag{15}$$

where $I_{\text{cond}}$ is a fixed conditioning tensor (e.g., reference and sketch) and $\mathbf{I}$ denotes the identity matrix. This parameterization allows the model to operate over a continuous spectrum of noise intensities, removing the timestep discretization bottleneck of DDPM [13].

**Noise-Aware Preconditioning**   To stabilize training and normalize feature magnitudes across varying $\sigma$, EDM applies a noise-aware preconditioning scheme [14]. The denoiser $D_\theta$ is constructed as a residual mapping composed of pre-scaled input/output paths:

$$D_\theta(I_{\text{noise}}, I_{\text{cond}}; \sigma) = c_{\text{skip}}(\sigma) I_{\text{noise}} + c_{\text{out}}(\sigma) \cdot F_\theta(c_{\text{in}}(\sigma) I_{\text{noise}}, I_{\text{cond}};\ c_{\text{noise}}(\sigma)), \tag{16}$$

where $c_\text{in}$, $c_\text{out}$, and $c_\text{skip}$ are scale-dependent coefficients defined as:

$$c_\text{skip} = \frac{\sigma_\text{data}^2}{\sigma^2 + \sigma_\text{data}^2}, \quad c_\text{out} = \frac{\sigma}{\sqrt{\sigma^2 + \sigma_\text{data}^2}}, \quad c_\text{in} = \frac{1}{\sqrt{\sigma^2 + \sigma_\text{data}^2}}, \quad c_\text{noise} = \frac{1}{4}\ln\sigma.$$

This formulation ensures that input features have consistent scale, preventing signal collapse at low noise or amplification at high noise levels. In practice, we use $\sigma_\text{data} = 0.5$.

**Training Objective**   Unlike DDPM which samples timesteps $t \in \{1, ..., T\}$, EDM samples $\ln\sigma$ from a normal distribution $\mathcal{N}(P_\text{mean}, P_\text{std}^2)$. The training loss is defined over random $\sigma$ as:

$$\mathcal{L} = \mathbb{E}_{\ln\sigma \sim \mathcal{N}(P_\text{mean}, P_\text{std}^2)} \mathbb{E}_{I_\text{gt} \sim p_\text{data}} \mathbb{E}_{\boldsymbol{n} \sim \mathcal{N}(0, \sigma^2\mathbf{I})} \left\| D_\theta(I_\text{gt} + \boldsymbol{n}, I_\text{cond}; \sigma) - I_\text{gt} \right\|^2. \tag{17}$$

**Sampling via Reverse-Time ODE**   At inference time, EDM uses a score-based formulation to define a reverse-time ordinary differential equation (ODE) that approximates the likelihood gradient with the denoiser output:

$$\nabla_{I_\text{noise}} \log p(I_\text{noise} \mid I_\text{cond}; \sigma) \approx \frac{D_\theta(I_\text{noise}, I_\text{cond}; \sigma) - I_\text{noise}}{\sigma^2}, \tag{18}$$

leading to the continuous reverse-time dynamics:

$$\frac{dI_\text{noise}}{dt} = -\frac{1}{\sigma}\left(D_\theta(I_\text{noise}, I_\text{cond}; \sigma) - I_\text{noise}\right). \tag{19}$$

**Sigma Schedule and Discretization**   To discretize this process, we apply the Euler method using a $\rho$-parameterized sigma schedule:

$$\sigma_i = \left[\sigma_\text{max}^{1/\rho} + \frac{i}{N-1}\left(\sigma_\text{min}^{1/\rho} - \sigma_\text{max}^{1/\rho}\right)\right]^\rho, \quad i = 0, 1, \dots, N-1. \tag{20}$$

We initialize the trajectory from pure noise $I^{(N-1)} \sim \mathcal{N}(0, \mathbf{I})$ and integrate the ODE in reverse over the precomputed $\{\sigma_i\}$ sequence. The denoising step at each index $i$ is performed as:

$$I^{(i-1)} = I^{(i)} - \frac{\Delta t_i}{\sigma_i}\left(D_\theta(I^{(i)}, I_\text{cond}; \sigma_i) - I^{(i)}\right), \quad \Delta t_i = \sigma_i - \sigma_{i-1}. \tag{21}$$

This continuous-time formulation enables [3] to benefit from the architectural and sampling improvements of EDM, while retaining its original conditioning and loss structure. In Section 4.1, we further extend this pipeline by introducing a perceptual scaling of $\sigma$ to ensure uniform SSIM degradation across steps.

## B   Details on SSIM-Aligned Sigma-Space Scaling

To design a perceptually uniform noise schedule, we empirically analyze the relationship between SSIM degradation and transformed noise levels $\phi(\sigma)$ across various candidate functions. For each transformation $\phi$, a clean image $I_\text{clean}$ is corrupted at $N = 50$ distinct noise levels by adding scaled Gaussian noise as described in (1). We then compute the SSIM between each noisy image and its clean counterpart to obtain a degradation curve. To quantify the perceptual consistency of each transformation, we plot SSIM values against $\phi(\sigma)$ and measure the linearity of the resulting curve using the coefficient of determination ($R^2$). This procedure is applied to 1% of randomly sampled training images, each undergoing 50 corruption steps, yielding a comprehensive perceptual degradation profile across a wide range of noise intensities.

As illustrated in Figure 4, plotting SSIM against $\phi(\sigma)$ reveals that certain transformations induce nearly linear degradation. In particular, bounded squash functions of the form

$$\phi(\sigma) = \frac{\sigma}{\sigma + c}$$

produce the most perceptually uniform trends. Among these, $\phi(\sigma) = \frac{\sigma}{\sigma+0.3}$ achieves near-perfect linearity with an $R^2$ value of 0.9949. Based on this result, we adopt this transformation as our default

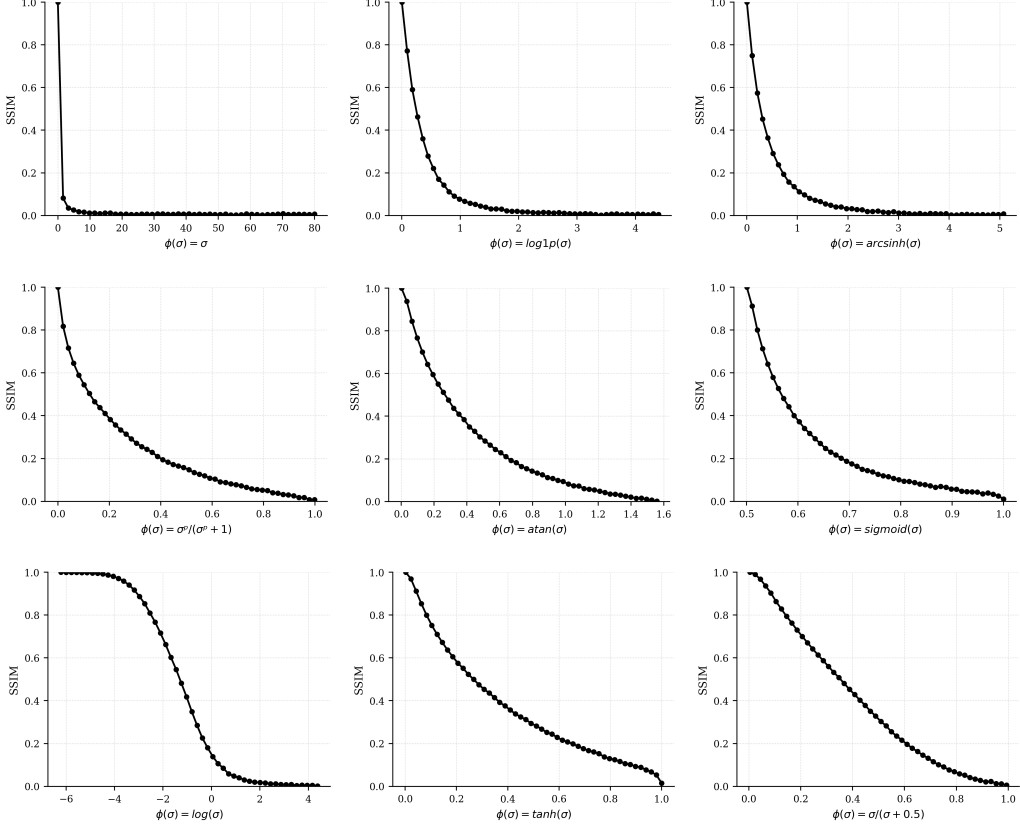

Figure 4: **SSIM degradation across various transformations** $\phi(\sigma)$. Each curve shows the SSIM between the clean image and its noisy counterpart as the noise level $\sigma$ increases, under a specific transformation $\phi$. The transformations are ordered by increasing linearity ($R^2$). Among them, bounded squash functions of the form $\phi(\sigma) = \frac{\sigma}{\sigma+c}$ exhibit the most linear degradation trends. In particular, $\phi(\sigma) = \frac{\sigma}{\sigma+0.3}$ achieves near-perfect linearity, making it well-suited for constructing perceptually uniform sigma schedules. For clarity, we visualize a representative subset of the evaluated transformations.

scaling function in sigma-space. Table 1 summarizes the $R^2$ values for representative candidate functions.

Finally, we construct our noise schedule by uniformly sampling steps in the transformed $\phi$-space and applying the inverse of the selected transformation to compute the corresponding $\sigma$ values, as defined in (5). This perceptually aligned schedule ensures that each diffusion step contributes uniformly to structural degradation, which is critical for achieving balanced and stable restoration during generation.

## C   Extended Qualitative Comparisons

To complement our main results, we present qualitative comparisons in both same-reference and cross-reference scenarios (Figures 5 and 6). In the same-reference scenario, our model produces visually faithful results that align well with both structure and style. In the cross-reference scenario, it generalizes robustly to unseen references, avoiding oversaturation and preserving content. These results highlight the benefit of SSIM-aligned sigma-space scaling and trajectory refinement in achieving perceptually consistent generation.

## C.1 Same-Reference Scenario

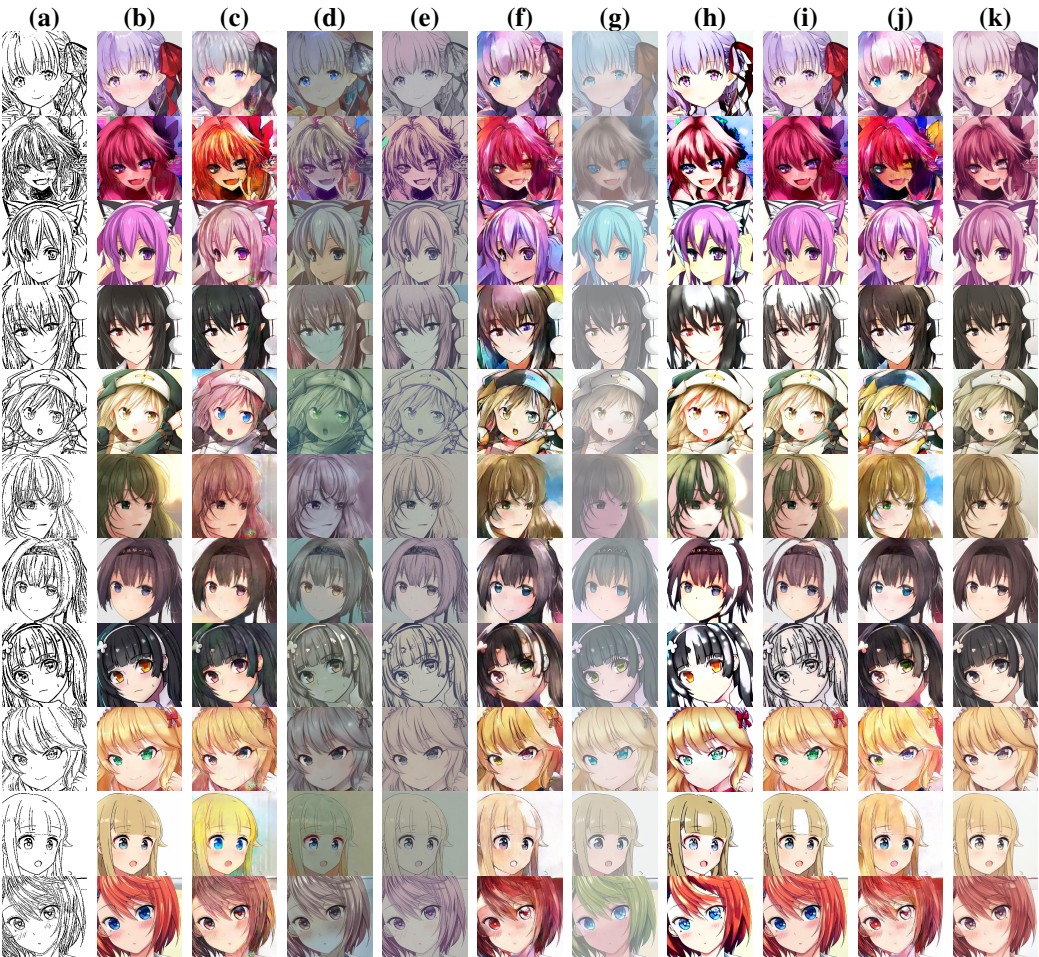

**(a)**    **(b)**    **(c)**    **(d)**    **(e)**    **(f)**    **(g)**    **(h)**    **(i)**    **(j)**    **(k)**

Figure 5: **Qualitative comparison under the same-reference scenario.** (a) Sketch input. (b) Reference image. (c) SCFT [18]. (d) AnimeDiffusion [3] (pretrained). (e) AnimeDiffusion [3] (finetuned). (f) AnimeDiffusion (EDM backbone, default $\sigma$-schedule). (g) ControlNet [36]. (h) Cross-Image Attention [1]. (i) Attention Distillation [39]. (j) Our model (w/o trajectory refinement). (k) Our model (w/ trajectory refinement).

## C.2 Cross-Reference Scenario

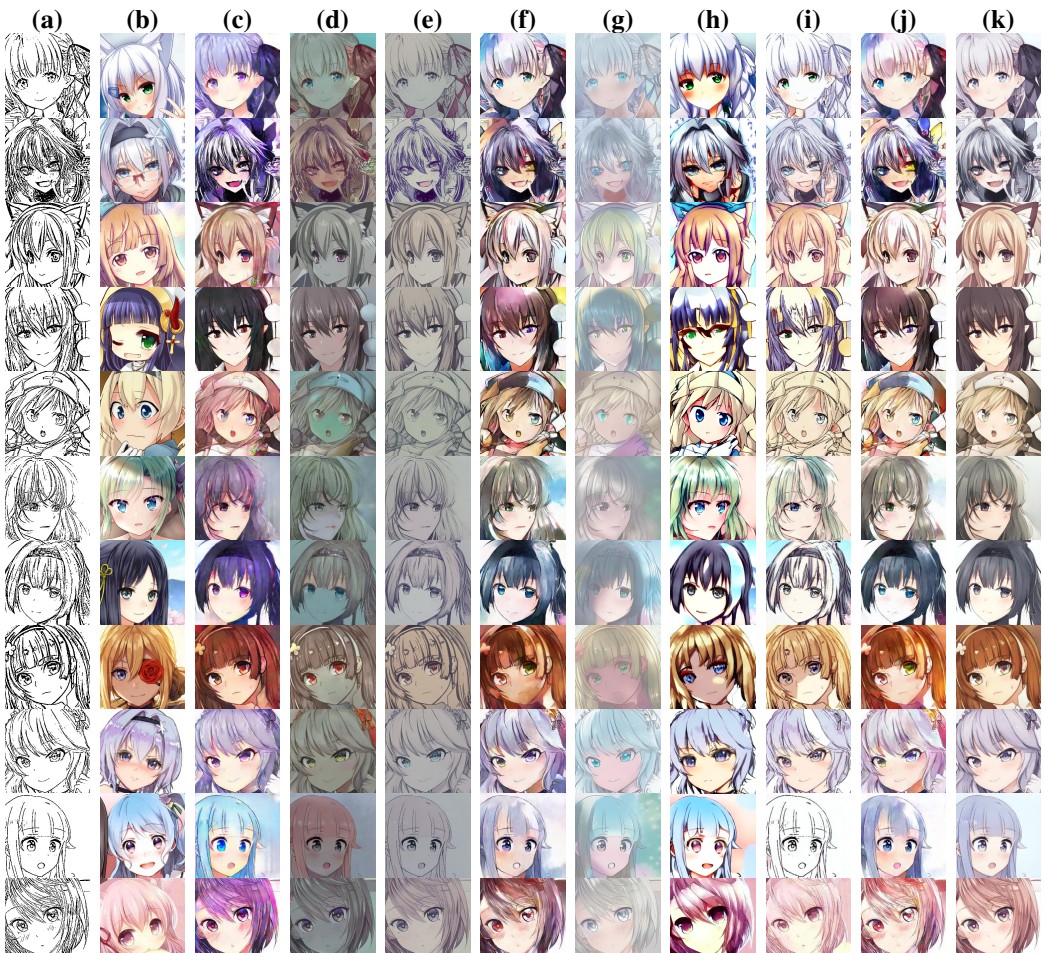

Figure 6: **Qualitative comparison under the cross-reference scenario.** (a) Sketch input. (b) Reference image. (c) SCFT [18]. (d) AnimeDiffusion [3] (pretrained). (e) AnimeDiffusion [3] (finetuned). (f) AnimeDiffusion (EDM backbone, default $\sigma$-schedule). (g) ControlNet [36]. (h) Cross-Image Attention [1]. (i) Attention Distillation [39]. (j) Our model (w/o trajectory refinement). (k) Our model (w/ trajectory refinement).

## D  Why Did We Add Rotation to TPS?

Table 4: Quantitative results without TPS rotation under both same-reference and cross-reference settings. Finetuning improves visual fidelity in both conditions.

| Method | PSNR ↑ | | MS-SSIM ↑ | | FID ↓ | |
|---|---|---|---|---|---|---|
| | Same | Cross | Same | Cross | Same | Cross |
| SSIMBaD (w/o trajectory refinement) | 20.55 | 11.34 | 0.8446 | 0.5996 | 56.18 | 65.69 |
| **SSIMBaD (w/ trajectory refinement)** | **23.10** | **14.00** | **0.9190** | **0.7714** | **24.35** | **40.73** |

Despite visually plausible results in Figure 8, especially after trajectory refinement, Table 4 reveals a significant performance gap between same- and cross-reference scenarios. For instance, PSNR drops from 23.10 dB to 14.00 dB, and MS-SSIM from 0.9190 to 0.7714, highlighting limited referential generalization. To address this, we introduce a lightweight affine rotation into the TPS pipeline, improving alignment between the sketch and reference. As shown in Table 2, incorporating TPS rotation reduces the PSNR and MS-SSIM gaps from 9.1 dB and 0.1476 to 3.08 dB and 0.0305, respectively. FID also improves, and our method surpasses all baselines under cross-reference scenario while retaining strong performance in the same-reference scenario.

## E  Does SSIM Behave as Intended During Generation?

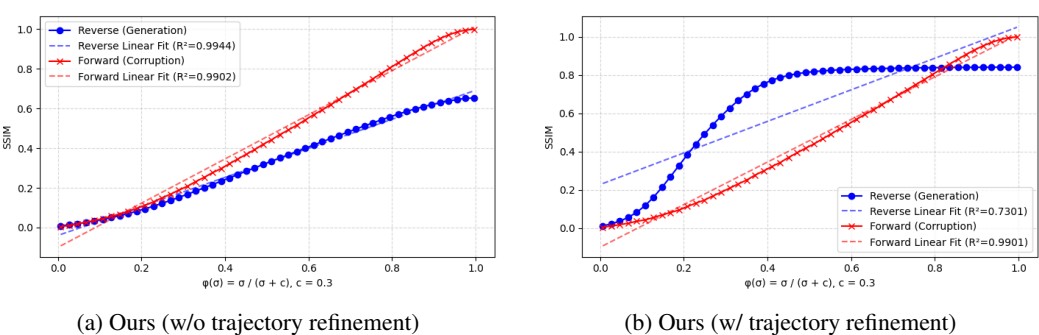

(a) Ours (w/o trajectory refinement)       (b) Ours (w/ trajectory refinement)

Figure 7: SSIM vs $\phi(\sigma)$ curves for the same input image under forward (corruption, red) and reverse (generation, blue) processes. Finetuning improves perceptual linearity in certain regions, but quickly saturates due to existing generation dynamics. The model nonetheless maintains an overall perceptually stable trajectory, suggesting potential for further improvement through trajectory-aware objectives.

To visually examine how closely the model's generation aligns with the intended noise schedule, we plot SSIM against $\phi(\sigma)^*$ for both the forward (corruption) and reverse (generation) processes, using the same input image and schedule.

Figure 7 compares this alignment before and after trajectory refinement. In both cases, the forward trajectory (red) shows near-perfect linear SSIM degradation, serving as a perceptual baseline. Notably, the reverse trajectory (blue) already exhibits a fair degree of linearity even before trajectory refinement, suggesting that the model implicitly learns to follow the $\phi(\sigma)^*$ path.

Importantly, trajectory refinement does not disrupt this linearity, preserving perceptual consistency while improving sample quality. These results highlight the robustness of our noise schedule and suggest that further improvements may be possible by designing more principled refinement objectives, which we leave for future work.

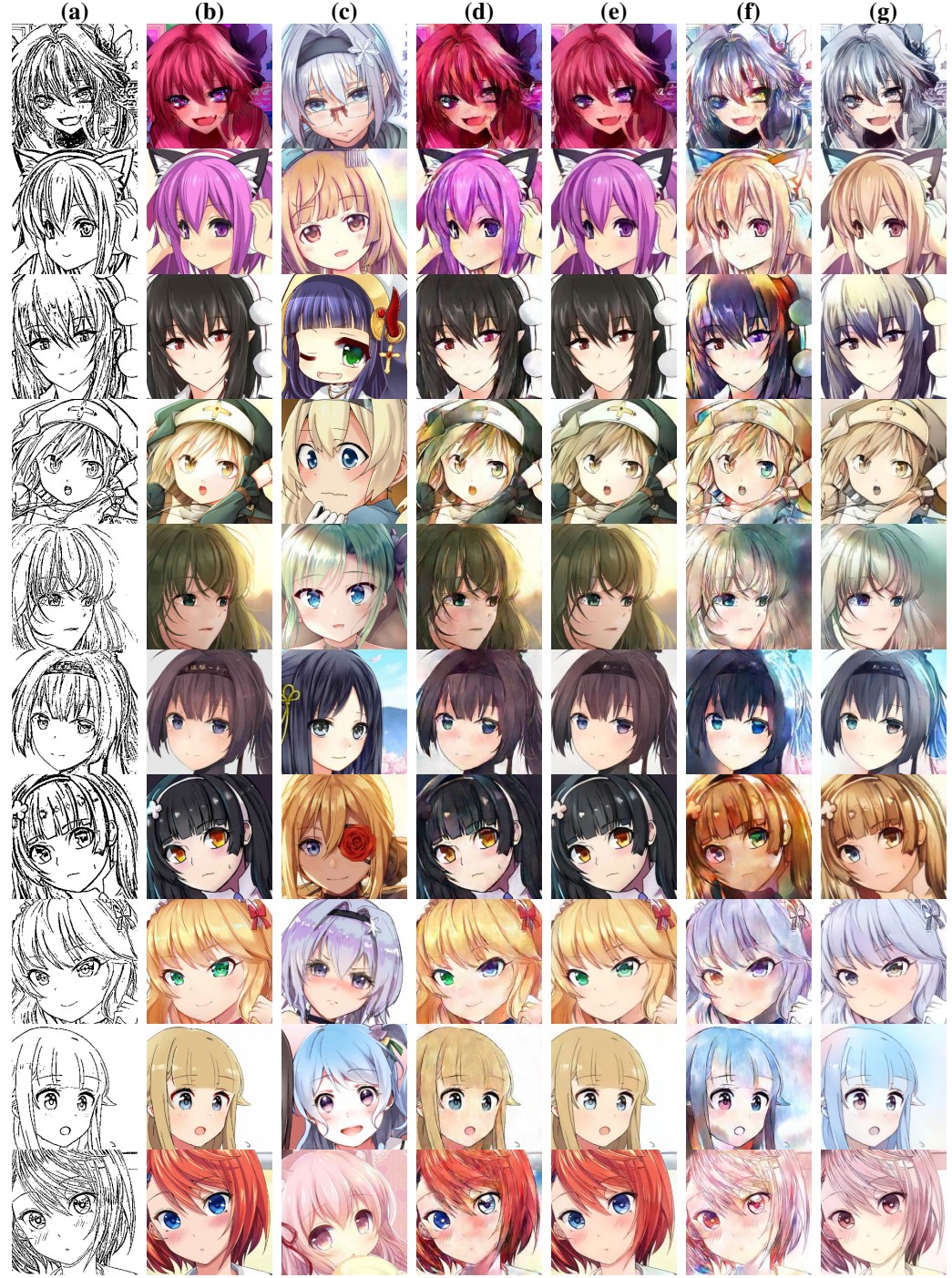

Figure 8: **Comparison under same- and cross-reference scenarios without TPS rotation.** (a) Sketch input. (b) Reference image (same style). (c) Reference image (cross style). (d–e) Our model under same-reference scenario (w/o and w/ trajectory refinement, no TPS rotation). (f–g) Our model under cross-reference scenario (w/o and w/ trajectory refinement, no TPS rotation). Even without explicit rotation-based alignment, our model preserves structural integrity and transfers style consistently across reference domains, outperforming baselines in both scenarios.

# F Implementation Details

To ensure rigorous and reproducible comparisons, we reimplemented each baseline model under a standardized pipeline. All models were trained and evaluated on the same dataset split, using identical image resolution ($256 \times 256$), batch size (32), and consistent data augmentation strategy.

**Hardware environment :** $2\times$ NVIDIA H100 SXM5 GPUs with a 128-core AMD EPYC 9354 CPU and 512GB RAM. Experiments were conducted using PyTorch 2.1.0 with AMP-based mixed-precision training.

**Common hyperparameters :**

- Optimizer: AdamW; Learning rate: $1 \times 10^{-4}$; Weight decay: $1 \times 10^{-2}$
- Scheduler: Cosine decay with 1 epoch warmup
- Epochs: 300; Batch size: 32; Gradient clipping: max-norm of 1.0
- Distributed training via PyTorch Lightning DDP; 64 data loading workers

## F.1 Pretraining Comparisons

For fair comparison of the **pretraining phase**, we evaluated models based on their ability to learn from distorted reference inputs and produce structure-preserving reconstructions.

**SCFT [18] :**

- Dense semantic correspondence-based reference transfer model originally designed for exemplar-guided colorization
- Adapted to $256 \times 256$ resolution
- Trained from scratch on our dataset with the same optimizer, learning rate schedule, and number of epochs

**AnimeDiffusion [3] :**

- Diffusion-based colorization model trained with fixed iDDPM-style $\beta$-schedule [13]
- Inference conducted using 50 denoising steps with DDIM [29]
- Official implementation modified for consistent data split and batch size

## F.2 Finetuning Comparisons

**Finetuning Settings :**

- Strategy: MSE, depending on baseline capability
- Inference time steps: 50 (Euler or DDIM sampling for diffusion models)
- Finetuning conducted with preloaded pretrained weights on the same hardware

**AnimeDiffusion [3] :**

- MSE-based perceptual finetuning with 50-step DDIM inference [29]
- Reference and sketch inputs preserved; distorted images created via noise+augmentation

**SSIMBaD (Ours):**

- Pretrained with SSIM-aligned $\phi^*(\sigma)$ schedule for uniform perceptual degradation
- Finetuned using MSE loss, with explicit control over 50 step inference trajectory

## F.3 Evaluation Metrics :

For both stages, we report PSNR, MS-SSIM [34], and FID [12]. All models were evaluated using 50-step sampling, and outputs were resized to $256 \times 256$ prior to metric computation.

