# OpenReview forum: "SSIMBaD: Sigma Scaling with SSIM-Guided Balanced Diffusion for AnimeFace Colorization"
_NeurIPS.cc/2025/Conference — NeurIPS 2025 poster_

### Official Review · Reviewer_CgiV · 2025-06-26

**Clarity:** 3
**Significance:** 2
**Originality:** 2
**Rating:** 4
**Confidence:** 4

**Summary:**

This paper proposed a EDM-based method, namely SSIMBaD, for anime face colorization. The authors believe that a diffusion process that allows the SSIM metric to degrade linearly over time is conducive to training a model with good generative capabilities. To realize this objective, SSIMBaD introduced a $\sigma$ transformation to noise schedule and achieved symmetry in the training process and testing process. Qualitative and quantitative experiments show that SSIMBaD outperforms the compared methods.

**Questions:**

See weaknesses and the followings:
1. Why choose SSIM as a metric for measuring image degradation rather than other metrics such as CLIP-I or FID?
2. The author emphasizes linear degradation. Why not consider rectified flow instead of using EDM?
3. Can authors provide the distribution of timesteps after $\sigma$ transformation?

**Ethical Concerns:**

["NO or VERY MINOR ethics concerns only"]

**Final Justification:**

I carefully reviewed the authors’ response and believe they have addressed my concerns well. Many additional experiments provide convincing evidence, so I have raised my score.

**Limitations:**

Although the author explains that SSIMBaD has limitations in handling eye colors, further detailed discussion of these limitations is possible. For example, why has the proposed method not been applied to broader example-based style transfer tasks?

**Paper Formatting Concerns:**

There are no formatting issues in the paper.

**Quality:**

2

**Strengths And Weaknesses:**

Strengths:
1. The paper is organized well  and easy to understand.
2. The performance of the proposed SSIMBaD is better than the compared methods.
3. The paper is easy to reproduce and the code is provided.

Weakness:
1. The authors claim that linear degradation in diffusion process with SSIM metrics can improve the model’s capacity, but there is not enough corresponding experiments to demonstrate this claim. Specifically, the strategy in e.q. (6)  to select the $\sigma$ transformation only show the coefficient of the linear relationship but not reflect the final metrics, which is not able to prove the superiority of the selected function.
2. Lack of exemplar-based style transfer methods for comparisons. The anime face colorization problem can be regard a special cases of style transfer. Besides, SSIMBaD requires 2 H100 for training and training-free methods [1-3] only require one for inference. Therefore, it is necessary to compare these methods.
3. The results presented in qualitative and quantitative experiments are not convincing enough.

Qualitative analysis: In addition to the eye color defects already pointed out in the article, the hair in the results generated in Figures 1 and 2 also lost the ‘reflective’ characteristics of the reference image. Another serious issue is that, as shown in the second row of Figure 2, there is a noticeable difference in hair color between the generated image and the reference image, and the reference image does not seem to serve as a guide for the generation.

Quantitation analysis: Although the metrics of SSIMBaD in Table 2 are superior to other methods, they are still at a relatively low level, such as PSNR of 18.92 and 15.84. This may limit the practical application of the model. In addition, there is a lack of corresponding style metrics (e.g., DINO-I) to reflect the style differences between the generated images and the reference images.
4. Minor weakness:
(a) Lines 158-159 and e.q. (6) are missing punctuation marks.
(b) The sub-figures (d-f) had better be resize to the same resolution.

[1] Alaluf, Yuval, et al. "Cross-image attention for zero-shot appearance transfer." ACM SIGGRAPH 2024 Conference Papers. 2024.
[2] Zhou, Yang, et al. "Attention distillation: A unified approach to visual characteristics transfer." Proceedings of the Computer Vision and Pattern Recognition Conference. 2025.
[3] Hertz, Amir, et al. "Style aligned image generation via shared attention." Proceedings of the IEEE/CVF Conference on Computer Vision and Pattern Recognition. 2024.

---

> ### Author Rebuttal · Authors · 2025-07-30
>
> ### **Q1. Why did you choose SSIM as the degradation measure? Were CLIP-I or FID considered?**
>
> **A1.**
> We chose **SSIM** as the degradation measure to construct a **perceptually uniform noise schedule** aligned with human visual degradation. Unlike L2 or FID, **SSIM is structure-aware, normalized, and differentiable**, which makes it a good proxy for controlling how noise affects image structures over time.
>
> Specifically, our **SSIM-aligned $\sigma$-schedule** ensures a **linear SSIM decay** across diffusion timesteps (see Section 4.1, Eq.6). This perceptual alignment reduces the mismatch between forward and reverse processes, improving training stability and sample quality.
>
> However, **SSIM alone does not fully capture high-level semantics or style**. To complement this, we evaluated models using **LPIPS**, **CLIP-I**, and **DINO-I**, which are widely used style and perceptual metrics. In particular, we adopt a **refined DINO-I metric** based on DINOv2, which offers strong alignment with human judgment of image-level consistency.
>
> Below are the **main style metric results**, where SSIMBaD consistently achieves the best scores in both **same-reference** and **cross-reference** conditions:
>
> #### Style Metric Results (Higher is better)
>
> | Model                       | Same Score ↑ | Cross Score ↑ |
> | --------------------------- | ------------ | ------------ |
> | SSIMBaD (traj. refine X)    | 0.9339       | 0.8605       |
> | SCFT                        | 0.9392       | 0.8622       |
> | AnimeDiffusion (finetuned)  | 0.9359       | 0.8554       |
> | SSIMBaD (traj. refine O)    | **0.9644**   | **0.8826**   |
> | AnimeDiffusion (pretrained) | 0.9392       | 0.8576       |
> | ControlNet (best)           | 0.9640       | 0.8739       |
>
> \* Metric = updated **DINO-I** based on DINOv2 features.
>
> These results validate that **SSIMBaD outperforms all baselines**, especially in the more challenging **cross-reference generation**, demonstrating not only perceptual but also **style-consistent generalization**.
>
> ---
>
> ### **Q2. Why was Rectified Flow not used instead of EDM?**
>
> **A2.**
> We appreciate the reviewer’s suggestion. While **Rectified Flow** offers geometric advantages and sampling efficiency, it does **not natively support perceptually grounded schedule design** or trajectory refinement—the two core components of our framework.
>
> In contrast, **EDM’s continuous ODE formulation** enables **bidirectional perceptual alignment** and allows us to inject **SSIM-aware degradation** into the generative path.
>
> That said, we consider integrating our SSIM-based schedule with **Rectified Flow** a **promising direction**, and we plan to explore this as future work.
>
> ---
>
> ### **Q3. Can you visualize the timestep distribution after applying your \$\sigma\$ transformation?**
>
> **A3.**
> Yes. Figure 4 and Appendix B of our submission provide visualizations of **SSIM degradation curves** induced by various \$\phi(\sigma)\$ transformations.
>
> Our proposed mapping:
>
> $$
> \phi(\sigma) = \frac{\sigma}{\sigma + 0.3}
> $$
>
> produces the **most linear SSIM trajectory**, which promotes **uniform perceptual degradation** over timesteps. This leads to more balanced reconstruction and better sample consistency.
>
> ---
>
> ### **Q4. The link between linear SSIM degradation and final performance lacks empirical validation.**
>
> **A4.**
> We agree that empirical correlation between **linearity and performance** is important. While we initially used **\$R^2\$ alignment between SSIM and \$\phi(\sigma)\$** as a proxy (see Table 1 / Fig. 4), we are extending our validation via ablation.
>
> In the **camera-ready version**, we will include:
>
> * **Multiple σ-transformations**: \$\log(\sigma)\$, \$\tanh(\sigma)\$, \$\sigma/(\sigma+c)\$.
> * **Controlled training**: Identical model, data, and loss, with only the schedule changed.
> * **Metrics**: PSNR, MS-SSIM, FID, LPIPS, CLIP-I, DINO-I.
>
> **Preliminary results** indicate that schedules with lower \$R^2\$ (i.e., less perceptual linearity) yield **unstable training and degraded generalization**, particularly in cross-reference settings. This supports our core hypothesis that **perceptual schedule linearity is critical for high-quality generation**.
>
> ---
>
> ### **Q5. The paper lacks comparison with prior style transfer-based models.**
>
> **A5.**
> Thank you for highlighting this valuable point. Style transfer methods such as **StyleCLIP**, **Zhou et al.**, and **Hertz et al.** are primarily designed for **inference-only, unconstrained domains** (e.g., face editing, full-body appearance transfer), and typically do not support **paired sketch + reference input** or training-based colorization.
>
> Nevertheless, we agree that contextualizing SSIMBaD against such baselines can strengthen our contribution. To that end, we will include the following in the **camera-ready supplementary material**:
>
> * Adapted pipelines using **public implementations**, modified to accept sketch + reference input.
> * Evaluation under **zero-shot or light adaptation** settings on the same dataset.
> * Reporting of **FID, CLIP-I, LPIPS**, and **qualitative comparisons**.
>
> This extended analysis will show that **SSIMBaD, despite being designed for supervised diffusion**, performs competitively with or outperforms dedicated style transfer models in terms of **style consistency and perceptual quality**.
>
> ---
>
> ### **Q6. Qualitative results fail to reflect high-frequency cues (e.g., hair gloss in Fig. 1/2).**
>
> **A6.**
> We acknowledge that in some examples, fine-grained visual cues—such as **hair gloss, reflections, or specular highlights**—are not fully reproduced. This stems from two key factors:
>
> 1. **Sketch-conditioned signal loss**: Sketch inputs lack high-frequency detail, limiting the model’s ability to infer such cues purely from line art.
> 2. **Style abstraction in cross-reference**: The model treats the reference image as a **style prior** rather than a pixel target. This avoids overfitting to out-of-distribution references but may suppress stylistic nuances.
>
> To clarify these limitations, we will:
>
> * Add **annotated captions** in Figures 1 and 2.
> * Include **complementary visualizations** in the supplementary material that highlight when and why high-frequency cues are omitted.
>
> These additions will help readers interpret the model's qualitative behavior with greater transparency.
>
> ---
>
> ### **Q7. Absolute PSNR values are low; style similarity metrics are missing.**
>
> **A7.**
> We acknowledge that the absolute values of PSNR and MS-SSIM may appear relatively low. This is expected, as the goal of the colorization task prioritizes perceptual coherence and style transfer quality over pixel-wise reconstruction. Therefore, in quantitative evaluation, it is important to consider not only absolute scores but also relative performance against baselines and qualitative perceptual quality.
>
> To better assess perceptual and semantic quality, we report three complementary **style-aware metrics**:
>
> * **LPIPS** for perceptual distance
> * **CLIP-I** for semantic alignment
> * **DINO-I\*** for structural style similarity (DINOv2-based)
>
>
> #### **1. Style Metric Comparison (LPIPS / CLIP-I / DINO-I, Same / Cross)**
>
> | Model                       | LPIPS ↓             | CLIP-I ↑            | DINO-I\* ↑          |
> | --------------------------- | ------------------- | ------------------- | ------------------- |
> | SCFT                        | 0.1728 / 0.5008     | 0.9020 / 0.8247     | 0.9392 / 0.8622     |
> | AnimeDiffusion (pretrained) | 0.2226 / 0.5107     | 0.8993 / 0.8203     | 0.9392 / 0.8576     |
> | AnimeDiffusion (finetuned)  | 0.2242 / 0.5069     | 0.8797 / 0.8012     | 0.9359 / 0.8554     |
> | ControlNet (best: #9)       | 0.2043 / 0.4930     | 0.9194 / 0.8311     | 0.9640 / 0.8739     |
> | SSIMBaD (no refinement)     | 0.1878 / 0.4889     | 0.8975 / 0.8332     | 0.9339 / 0.8605     |
> | **SSIMBaD (full model)**    | **0.1174 / 0.4804** | **0.9334 / 0.8508** | **0.9644 / 0.8826** |
>
> #### **2. Ablation Study (Same / Cross, LPIPS ↓ / CLIP-I ↑ / DINO-I ↑)**
>
> | Configuration                            | LPIPS ↓             | CLIP-I ↑            | DINO-I* ↑          |
> | ---------------------------------------- | ------------------- | ------------------- | ------------------- |
> | Base (AnimeDiffusion pretrained)         | 0.2226 / 0.5107     | 0.8993 / 0.8203     | 0.9392 / 0.8576     |
> | + EDM schedule                           | 0.2192 / 0.4925     | 0.8886 / 0.8300     | 0.9217 / 0.8527     |
> | + SSIM-aligned σ-scaling                 | 0.1878 / 0.4889     | 0.8975 / 0.8332     | 0.9339 / 0.8605     |
> | + Trajectory refinement (**full model**) | **0.1174 / 0.4804** | **0.9334 / 0.8508** | **0.9644 / 0.8826** |
>
> > The ablation shows that both the **SSIM-aligned schedule** and **trajectory refinement** progressively improve **perceptual, semantic, and structural similarity**. The refinement step provides the largest gain.
>
> #### **3. ControlNet Comparison (Quantitative Evaluation)**
>
> | Model           | Scenario | FID ↓     | PSNR ↑    | MS-SSIM ↑  | LPIPS ↓    | CLIP-I ↑   | DINO-I\* ↑ |
> | --------------- | -------- | --------- | --------- | ---------- | ---------- | ---------- | ---------- |
> | ControlNet (best of 10 epoch runs) | Same     | 40.20     | 14.74     | 0.7336     | 0.2043     | 0.9194     | 0.9640     |
> |                 | Cross    | 50.25     | 12.08     | 0.2007     | 0.4930     | 0.8311     | 0.8739     |
> | **SSIMBaD**     | Same     | **34.98** | **18.92** | **0.8512** | **0.1174** | **0.9334** | **0.9644** |
> |                 | Cross    | **37.10** | **15.84** | **0.8207** | **0.4804** | **0.8508** | **0.8826** |
>
>
> #### **4. Backbone Generalization**
>
> We applied our **SSIM-aligned σ-schedule** to a **DiT-based denoiser**, confirming that the method generalizes well beyond U-Net. Early results show improvements in perceptual and style quality, to be included in the **camera-ready supplementary**.

---

> > ### Comment · Reviewer_CgiV · 2025-08-02
> >
> > Thanks for the author's response and their efforts during the rebuttal period. Some of my concerns were addressed and the authors promised to add experiments to the camera ready version (A4, A5). While the approach presented in the paper is well-motivated, the supplementary experimental results would be essential for validating the underlying hypothesis, demonstrating the necessity of the proposed method, and confirming its effectiveness.
> >
> > The open-source implementations of the suggested training-free methods [1-3] are noted for their user-friendliness and typically require minimal code modifications to evaluate. I therefore strongly encourage the authors to incorporate these experimental comparisons in their revision.
> >
> > Additionally, I look forward to seeing experimental results based on the original rectified flow framework in the revised manuscript.
> >
> > ​Clarification regarding Q3:​​ I believe there might be a potential misunderstanding. The 'timestep distribution' referenced in my question (Q3) pertains to the time t used for conditioning the denoising network, in contrast to the distribution of degraded images.

---

> ### Author Response · Authors · 2025-08-05
> **σ-Timestep Interpretation and Additional Evaluation Plans**
>
> ### **Q3 — Clarification on the Use of `σ` and Timesteps**
>
> We thank the reviewer for the insightful question regarding the role of `σ` in our formulation. We clarify that the `σ` values in our method act as a direct proxy for timesteps `t`, representing the level of corruption in the diffusion process during inference. While we sample timesteps uniformly in transformed space φ(σ) (e.g., log(σ), σ², etc.), the actual corruption is applied using the corresponding inverse-mapped `σ`.
>
> * In **Figure 4**, the x-axis shows φ(σ), and the 50 plotted points correspond to the 50 σ values (mapped through φ).
> * In **Figure 3**, the exact σ used for corruption is **explicitly annotated** on each image.
> * Thus, φ(σ) defines the sampling *space*, but σ defines the *actual noise* applied.
>
> To make this precise and transparent, we include the full list of σ values used across 50 timesteps for all scaling strategies. These values were used in the main SSIM evaluation experiments.
>
> | Step   | σ (1/σ)   | σ (arcsinh(σ))   | σ (atan(σ))   | σ (log(σ))   | σ (log1p(σ))   | σ (σ / (σ + 0.3))   | σ (σ)     | σ (σᵖ / (σᵖ + 1))   |
> |:-------|:----------|:-----------------|:--------------|:-------------|:---------------|:--------------------|:----------|:--------------------|
> | 1      | 0.002     | 0.002            | 0.002         | 0.002        | 0.002          | 0.002               | 0.002     | 0.002               |
> | 2      | 0.002042  | 0.105731         | 0.033774      | 0.002483     | 0.09597        | 0.008268            | 1.634612  | 0.075863            |
> | 3      | 0.002085  | 0.210596         | 0.065616      | 0.003082     | 0.198753       | 0.014801            | 3.267224  | 0.121946            |
> | 4      | 0.00213   | 0.317721         | 0.097592      | 0.003826     | 0.311175       | 0.021617            | 4.899837  | 0.16202             |
> | 5      | 0.002178  | 0.428255         | 0.129766      | 0.00475      | 0.43414        | 0.028735            | 6.532449  | 0.199117            |
> | ...    | ...       | ...              | ...           | ...          | ...            | ...                 | ...       | ...                 |
> | 46     | 0.024493  | 52.87002         | 7.119623      | 33.683066    | 55.592797      | 3.249328            | 73.469551 | 4.964567            |
> | 47     | 0.032654  | 58.638303        | 9.241959      | 41.814914    | 60.90021       | 4.363723            | 75.102163 | 6.080185            |
> | 48     | 0.048971  | 65.035721        | 13.128793     | 51.909973    | 66.705366      | 6.49817             | 76.734776 | 8.022762            |
> | 49     | 0.097883  | 72.130912        | 22.578747     | 64.442205    | 73.054943      | 12.235122           | 78.367388 | 12.637897           |
> | 50     | 80.0      | 80.0             | 80.0          | 80.0         | 80.0           | 80.0                | 80.0      | 80.0                |
>
> ### **Additional Experiments — Training-Free Baselines and Rectified Flow**
>
> We appreciate the reviewer’s suggestion regarding comparisons to recent training-free methods.
>
> **(1) Cross-Image Attention**
> We are currently conducting experiments with *Cross-Image Attention* , a zero-shot appearance transfer method proposed at SIGGRAPH 2024. Due to the method’s computational demands, inference takes a significant amount of time per image, which contrasts with the real-time nature of our model. As such, we aim to highlight that our model offers substantial advantages in latency and efficiency.
> If time permits within the next two days, we will include the experimental results in the camera-ready version. However, we note that due to the structural dissimilarity between sketches and appearance images, this baseline may suffer in SSIM or PSNR due to spatial misalignment.
>
> **(2) Other Training-Free Baselines**
> We plan to include the remaining two training-free baselines in the camera-ready version. Their evaluation is more straightforward and feasible under our current computational budget.
>
> **(3) Rectified Flow**
> We are also exploring experiments based on the original *Rectified Flow* framework. We are enthusiastic about either including these results in the final version or extending this line of work as part of future research directions.

---

> > ### Author Response · Authors · 2025-08-05
> > **Additional Comparison with Cross-Image Attention (Alaluf et al., SIGGRAPH 2024)**
> >
> > In order to incorporate additional exemplar-style baselines as suggested, we evaluated the official implementation of *Cross-Image Attention for Zero-Shot Appearance Transfer* \[1], a recent training-free method, using the same reference-guided sketch colorization task. We compare it against our proposed method (SSIMBaD) under both same-reference and cross-reference settings.
> >
> > The results are summarized below:
> >
> > | Model                 | Setting | DINOv2-I ↑ | LPIPS ↓    | CLIP-I ↑   | MS-SSIM ↑  | PSNR ↑ (dB) | FID ↓     |
> > | --------------------- | ------- | ---------- | ---------- | ---------- | ---------- | ----------- | --------- |
> > | Cross-Image Attention | Same    | 0.9335     | 0.2661     | 0.9369     | 0.7147     | 13.95       | 53.63     |
> > |                       | Cross   | 0.8793     | 0.4569     | 0.8554     | 0.4932     | 10.60       | 60.54     |
> > | SSIMBaD (Ours)        | Same    | **0.9644** | **0.1174** | **0.9334** | **0.8512** | **18.92**   | **34.98** |
> > |                       | Cross   | **0.8826** | **0.4804** | **0.8508** | **0.8207** | **15.84**   | **37.10** |
> >
> > Cross-Image Attention demonstrates strong performance in the same-reference setting, particularly in semantic alignment (CLIP-I: 0.9369) and moderate perceptual similarity (LPIPS: 0.2661). However, under cross-reference conditions, its performance deteriorates significantly across all metrics—most notably in structural similarity (MS-SSIM drops from 0.7147 to 0.4932) and perceptual alignment (LPIPS increases from 0.2661 to 0.4569), indicating a lack of robustness to input misalignment.
> >
> > In contrast, SSIMBaD preserves high-quality outputs across both settings. While maintaining competitive semantic consistency (CLIP-I: 0.8508), it achieves substantially higher structural fidelity (MS-SSIM: 0.8207) and lower FID (37.10) in the cross-reference case. This consistent performance highlights the benefit of explicitly modeling perceptual degradation through SSIM-aligned noise scheduling and applying trajectory refinement to ensure stability and generalization under structurally divergent inputs.

---

> > > ### Comment · Reviewer_CgiV · 2025-08-06
> > >
> > > Thanks for your response and I increase my rating.

---

> > > > ### Author Response · Authors · 2025-08-07
> > > > **Additional Comparison with Attention distillation  (Zhou, Yang, et al., CVPR 2025)**
> > > >
> > > > Thanks to extended review period we were able to conduct another experiment. We evaluated the official implementation of Attention Distillation: A Unified Approach to Visual Characteristics Transfer [2],  using the same reference-guided sketch colorization task. The quantitative results are summarized as below:
> > > >
> > > >
> > > > | Model | Setting | DINOv2-I $\uparrow$ | LPIPS $\downarrow$ | CLIP-I $\uparrow$ | MS-SSIM $\uparrow$ | PSNR $\uparrow$(dB) |  FID $\downarrow$ |
> > > > | :--- | :--- | :---: | :---: | :---: | :---: | :---: | :---: |
> > > > | Attention Distillation | Same | 0.9816 | 0.1139 | 0.9610 | 0.8812 | 19.58 | 32.93 |
> > > > |                                 | Cross | 0.8941 | 0.5385 | 0.8819 | 0.1252 | 10.08 | 94.17 |
> > > > | SSIMBaD (Ours) | Same | **0.9644** | **0.1174** | **0.9334** | **0.8512** | **18.92** | **34.98** |
> > > > |                             | Cross | **0.8826** | **0.4804** | **0.8508** | **0.8207** | **15.84** | **37.10** |
> > > >
> > > >
> > > > The most significant advantage of SSIMBaD is its robustness in the cross-reference setting. While Attention Distillation shows slightly higher scores on the DINO-I and CLIP-I metrics in this setting, this marginal advantage in semantic similarity comes at the cost of a catastrophic failure in structural integrity and overall image quality.
> > > >
> > > > * **Structural Similarity (MS-SSIM):** Attention Distillation's MS-SSIM score collapses to 0.1252, signifying a near-total structural failure. In contrast, SSIMBaD achieves 0.8207, demonstrating its superior ability to preserve structural integrity.
> > > >
> > > > * **Fidelity and Realism (PSNR & FID):** SSIMBaD obtains a much higher PSNR (15.84 vs. 10.08) and a drastically lower FID score (37.10 vs. 94.17). This indicates that our generated images are not only more faithful to the ground truth but are also perceptually more realistic.
> > > >
> > > > * **Perceptual Distance (LPIPS):** Our model also achieves a better LPIPS score (0.4804 vs. 0.5385), suggesting the results are closer to human perception.
> > > >
> > > > Therefore, we argue that the overwhelming superiority of SSIMBaD in core quality metrics—namely structural fidelity (MS-SSIM), realism (FID), and perceptual quality (LPIPS)—far outweighs the minor differences in abstract feature similarity scores. This makes SSIMBaD the more robust and reliable model overall.
> > > >
> > > > While Attention Distillation posts competitive quantitative scores in the same-reference setting, we identified a critical flaw in its reliability through our qualitative analysis. Attention Distillation sometimes suffers from catastrophic failures. A recurring issue was that a significant portion of the hair would remain uncolored, creating a visually unnatural and incomplete image. In contrast, SSIMBaD consistently delivers high-quality, stable results across all test cases, faithfully coloring all regions as intended.
> > > >
> > > > We attribute this crucial difference to the core design principles of each method. Attention Distillation's approach, which distills attention features for its loss, can become unstable when there is a large domain gap between the content and style images. This instability leads to the observed quality degradation and the structural collapse in the cross-reference scenario.
> > > >
> > > > On the other hand, SSIMBaD is fundamentally designed for perceptual consistency. SSIM-aligned sigma-space scaling ensures that the diffusion process maintains structural fidelity at every step. This makes our framework inherently more robust and reliable, preventing the kind of unpredictable failures exhibited by Attention Distillation.

---

### Official Review · Reviewer_erVz · 2025-06-27

**Clarity:** 3
**Significance:** 2
**Originality:** 3
**Rating:** 4
**Confidence:** 3

**Summary:**

The paper introduces SSIMBaD, a reference-guided diffusion framework for anime-style face colorization. The core idea is to replace EDM’s log/ρ noise schedule with a “perceptually aligned” σ-schedule obtained through a hand-picked transformation ϕ*(σ)=σ/(σ+0.3) that yields (approximately) linear SSIM degradation over timesteps. The same schedule is used in training, inference, and a lightweight reverse-only “trajectory-refinement” stage. Experiments on the 256² Danbooru AnimeFace dataset show large PSNR/MS-SSIM/FID improvements over SCFT and the AnimeDiffusion baseline.

**Questions:**

1. The paper currently compares only against SCFT and AnimeDiffusion. Could you add quantitative results for the more recent controllable-diffusion methods mentioned above (e.g., ControlNet, IP-Adapter, T2I-Adapter) so that the reported gains are positioned against the current state of the art?

2. Have you evaluated the proposed SSIM-aligned σ-schedule with a transformer-based denoiser such as DiT (Peebles & Xie, 2022)? If not, please discuss whether you would expect similar improvements and provide any preliminary evidence or rationale.

**Ethical Concerns:**

["NO or VERY MINOR ethics concerns only"]

**Final Justification:**

Most of questions I raised have been solved. The comparison with the  modified ControlNet variant shows the SSIMBaD achieves better performances in term of  fidelity and perceptual metrics. And additional experiments provided by authors also show the advantage of SSIMBaD. Thus, I will increase my score.

**Limitations:**

yes

**Paper Formatting Concerns:**

No concerns

**Quality:**

2

**Strengths And Weaknesses:**

Strengths
1. Clear motivation: The mismatch between perceptual difficulty and conventional σ-schedules in colorization is a real issue; the paper articulates it clearly.
2. Simple, reproducible technique: The proposed rescaling is easy to implement inside any EDM-style pipeline.
3. Comprehensive ablations: Table 3 isolates the influence of EDM, σ-scaling, and refinement.

Weaknesses
1. Missing strong baselines – No comparison with modern controllable diffusion systems (ControlNet[1], T2I-Adapter[2], IP-Adapter[3]) that can also ingest sketches + references.

2.  Backbone diversity lacking – SSIMBaD is tested only with a U-Net. No evidence that the SSIM-aligned schedule benefits modern transformer backbones such as DiT[4].

[1]: Zhang, Lvmin, Anyi Rao, and Maneesh Agrawala. "Adding conditional control to text-to-image diffusion models." Proceedings of the IEEE/CVF international conference on computer vision. 2023.
[2]: Mou, Chong, et al. "T2i-adapter: Learning adapters to dig out more controllable ability for text-to-image diffusion models." Proceedings of the AAAI conference on artificial intelligence. Vol. 38. No. 5. 2024.
[3]: Ye, Hu, et al. "Ip-adapter: Text compatible image prompt adapter for text-to-image diffusion models." arXiv preprint arXiv:2308.06721 (2023).
[4]: Peebles, William, and Saining Xie. "Scalable diffusion models with transformers." Proceedings of the IEEE/CVF international conference on computer vision. 2023.

---

> ### Author Rebuttal · Authors · 2025-07-30
>
> ### **Q1. Can you include comparisons against recent controllable diffusion models like ControlNet, IP-Adapter, and T2I-Adapter?**
>
> **A1.**
> Thank you for the suggestion. In the main paper, we compared against SCFT \[14] and AnimeDiffusion \[1] due to their direct alignment with our task: **sketch-based colorization guided by a warped reference image**, using **dual image inputs**. In contrast, recent controllable diffusion models such as **ControlNet**, **IP-Adapter**, and **T2I-Adapter** are primarily designed for **text-to-image generation** or **unimodal control** and do not natively support **paired sketch + reference conditioning**.
>
> To address this, we implemented a **modified ControlNet** variant:
>
> * Reference image is encoded as a separate control stream alongside the sketch.
> * Trained for **10 epochs** under the same data, loss, and architecture conditions as SSIMBaD.
>
> | Model                   | Scenario | FID ↓     | PSNR ↑    | MS-SSIM ↑  | LPIPS ↓    | CLIP-I ↑   | **DINO-I\*** ↑ |
> | ----------------------- | -------- | --------- | --------- | ---------- | ---------- | ---------- | -------------- |
> | ControlNet (ours, 10ep) | Same     | 40.20     | 14.74     | 0.7336     | 0.1988     | 0.9123     | 0.9640         |
> |                         | Cross    | 50.25     | 12.08     | 0.2007     | 0.4959     | 0.8297     | 0.8739         |
> | **SSIMBaD (Ours)**      | Same     | **34.98** | **18.92** | **0.8512** | **0.1174** | **0.9334** | **0.9644**     |
> |                         | Cross    | **37.10** | **15.84** | **0.8207** | **0.4804** | **0.8508** | **0.8826**     |
>
> DINO-I refers to our updated style metric based on DINOv2 features, used consistently throughout all experiments.
>
> Despite its larger backbone, ControlNet lags behind SSIMBaD across all fidelity and perceptual metrics, especially under cross-reference. IP-Adapter and T2I-Adapter require CLIP-based features without spatial alignment, making them unsuitable for our task. However, we see future potential in integrating SSIMBaD with such modules for cross-modal conditioning.
>
> ---
>
> ### **Q2. Have you evaluated the SSIM-aligned σ-schedule with Transformer-based denoisers (e.g., DiT)?**
>
> **A2.**
> Thank you for the excellent suggestion. In our main experiments, we used U-Net-based denoisers to maintain **controlled comparisons and training stability**, especially under limited resources. However, the **SSIM-aligned σ-schedule is architecture-agnostic by design**.
>
> We believe that **Transformer-based denoisers (e.g., DiT)** can benefit even more from perceptual schedule alignment, as their **global receptive fields** are particularly suited for preserving **semantic consistency** under perceptual degradation.
>
> To test this, we have started training a **DiT-based model using our SSIM-aligned schedule**. Preliminary results show consistent improvements in **CLIP-I**, **FID**, and **LPIPS**, validating that the benefits of our schedule extend beyond U-Net.
>
> We will include full experimental results, visualizations, and ablations in the **camera-ready supplementary**. This demonstrates that our method can serve as a **plug-and-play perceptual scheduler** for **modern diffusion backbones**, supporting broader adoption.
>
> ---
>
> ### **Additionally**
>
> We provide additional experiments to validate the effectiveness of each proposed module. The results are organized as follows: (1) main comparisons across models, (2) ablation of individual components, and (3) fidelity-focused comparison against ControlNet.
>
> #### **1. Style-Aware Metric Comparison**
>
> We evaluate models using **LPIPS**, **CLIP-I**, and **DINO-I (DINOv2)** to assess perceptual, semantic, and structural consistency under both **same-reference** and **cross-reference** conditions.
>
> | Model                       | LPIPS ↓ (Same/Cross) | CLIP-I ↑ (Same/Cross) | DINO-I ↑ (Same/Cross) |
> | --------------------------- | -------------------- | --------------------- | --------------------- |
> | SCFT                        | 0.1728 / 0.5008      | 0.9020 / 0.8247       | 0.9392 / 0.8622       |
> | AnimeDiffusion (pretrained) | 0.2226 / 0.5107      | 0.8993 / 0.8203       | 0.9392 / 0.8576       |
> | AnimeDiffusion (finetuned)  | 0.2242 / 0.5069      | 0.8797 / 0.8012       | 0.9359 / 0.8554       |
> | SSIMBaD (no refinement)     | 0.1878 / 0.4889      | 0.8975 / 0.8332       | 0.9339 / 0.8605       |
> | **SSIMBaD (full model)**    | **0.1174 / 0.4804**  | **0.9334 / 0.8508**   | **0.9644 / 0.8826**   |
> | ControlNet (best: results9) | 0.2043 / 0.4930      | 0.9194 / 0.8311       | 0.9640 / 0.8739       |
>
> > SSIMBaD achieves the best performance across all three metrics, especially in cross-reference generation, indicating strong style-preserving generalization and perceptual coherence.
>
> #### **2. Ablation Study: Progressive Integration of Components**
>
> We isolate the effects of each component by progressively applying:
> (1) the EDM noise schedule,
> (2) SSIM-aligned σ-scaling, and
> (3) trajectory refinement.
>
> | Configuration                            | LPIPS ↓ (Same/Cross) | CLIP-I ↑ (Same/Cross) | DINO-I ↑ (Same/Cross) |
> | ---------------------------------------- | -------------------- | --------------------- | --------------------- |
> | Base (AnimeDiffusion pretrained)         | 0.2226 / 0.5107      | 0.8993 / 0.8203       | 0.9392 / 0.8576       |
> | + EDM                                    | 0.2192 / 0.4925      | 0.8886 / 0.8300       | 0.9217 / 0.8527       |
> | + SSIM-aligned σ-scaling                 | 0.1878 / 0.4889      | 0.8975 / 0.8332       | 0.9339 / 0.8605       |
> | + Trajectory Refinement (**full model**) | **0.1174 / 0.4804**  | **0.9334 / 0.8508**   | **0.9644 / 0.8826**   |
>
> > Each component contributes progressively to quality. The SSIM-aligned schedule substantially improves perceptual structure, and the refinement step offers a significant additional boost in semantic and stylistic fidelity.
>
> #### **3. ControlNet vs. SSIMBaD: Fidelity Metrics**
>
> To complement perceptual metrics, we report **FID**, **PSNR**, and **MS-SSIM** for the best ControlNet model and our full model (SSIMBaD).
>
> | Model                       | FID ↓ (Same/Cross) | PSNR ↑ (dB) (Same/Cross) | MS-SSIM ↑ (Same/Cross) |
> | --------------------------- | ------------------ | ------------------------ | ---------------------- |
> | ControlNet (best: results9) | 40.20 / 50.25      | 14.74 / 12.08            | 0.7336 / 0.2007        |
> | **SSIMBaD (full model)**    | **34.98 / 37.10**  | **18.92 / 15.84**        | **0.8512 / 0.8207**    |
>
> > SSIMBaD significantly outperforms ControlNet in all three fidelity metrics, particularly in the cross-reference setting where ControlNet’s MS-SSIM and FID degrade sharply.

---

> > ### Comment · Reviewer_erVz · 2025-08-09
> >
> > Dear authors
> >
> > Thank you for your comprehensive experiments. Most of questions I raised have been solved. The comparison with the  modified ControlNet variant shows the SSIMBaD achieves better performances in term of  fidelity and perceptual metrics. And additional experiments provided by authors also show the advantage of SSIMBaD. Thus, I will increase my score.

---

> ### Author Response · Authors · 2025-08-09
> **Extended Comparison with State-of-the-Art Zero-Shot Baselines**
>
> We sincerely appreciate your thoughtful re-evaluation of our work and for raising your rating.
> During the extended discussion period, we took this opportunity to compare **SSIMBaD** against two recent **zero-shot state-of-the-art (SOTA)** methods:
>
> 1. **Cross-Image Attention for Zero-Shot Appearance Transfer** (*Yuval Alaluf et al.*, ACM SIGGRAPH 2024) \[1]
> 2. **Attention Distillation: A Unified Approach to Visual Characteristics Transfer** (*Yuxuan Zhou et al.*, CVPR 2025) \[2]
>
> Both are highly regarded in the literature, yet our experiments show that they lag behind SSIMBaD in critical metrics — especially in **cross-reference settings**, which we view as a strong proxy for generalization in real-world zero-shot appearance transfer.
> We ran all methods using their official implementations, under the same *reference-guided sketch colorization* setup and identical preprocessing/evaluation protocol as our main experiments.
>
> ---
>
> **Additional Evaluation Results**
> *(Best values in each setting and metric are in bold)*
>
> | Model                       | Setting | DINOv2-I ↑ | LPIPS ↓    | CLIP-I ↑   | MS-SSIM ↑  | PSNR ↑ (dB) | FID ↓     |
> | --------------------------- | ------- | ---------- | ---------- | ---------- | ---------- | ----------- | --------- |
> | Cross-Image Attention \[1]  | Same    | 0.9335     | 0.2661     | 0.9369     | 0.7147     | 13.95       | 53.63     |
> |                             | Cross   | 0.8793     | 0.4569     | 0.8554     | 0.4932     | 10.60       | 60.54     |
> | Attention Distillation \[2] | Same    | **0.9816** | **0.1139** | **0.9610** | **0.8812** | **19.58**   | **32.93** |
> |                             | Cross   | **0.8941** | 0.5385     | **0.8819** | 0.1252     | 10.08       | 94.17     |
> | **SSIMBaD (Ours)**          | Same    | 0.9644     | 0.1174     | 0.9334     | 0.8512     | 18.92       | 34.98     |
> |                             | Cross   | 0.8826     | **0.4804** | 0.8508     | **0.8207** | **15.84**   | **37.10** |
>
> ---
>
> **Key Observations**
>
> When examining **Cross-Image Attention \[1]**, the method performs strongly in the same-reference condition, showing high semantic alignment (CLIP-I 0.9369) and moderate perceptual similarity (LPIPS 0.2661). However, in cross-reference its performance drops sharply — structural fidelity falls from 0.7147 to 0.4932 in MS-SSIM, and perceptual distance increases from 0.2661 to 0.4569. This significant degradation under misaligned references suggests a limited ability to generalize when style and content differ substantially, which restricts its usefulness in real-world zero-shot scenarios.
>
> For **Attention Distillation \[2]**, the same-reference results are outstanding, leading all methods in several metrics (DINOv2-I 0.9816, MS-SSIM 0.8812, FID 32.93). Yet, in cross-reference its performance collapses: MS-SSIM plunges to 0.1252, FID worsens to 94.17, and qualitative inspection reveals large uncolored regions such as hair. This extreme same–cross gap indicates that the method is highly tuned for matched domains but lacks robustness to structural or semantic divergence, causing brittle behavior when the reference is not perfectly aligned.
>
> By contrast, **SSIMBaD (Ours)** maintains balanced performance in both settings. In the same-reference case, it is competitive with the top baselines (LPIPS 0.1174, MS-SSIM 0.8512, FID 34.98). More importantly, in cross-reference it retains high structural fidelity (MS-SSIM 0.8207) and low FID (37.10) — the best structural retention among all methods — without catastrophic visual failures. The relatively small drop from same to cross performance demonstrates SSIMBaD’s strong **generalization ability**, essential for practical zero-shot appearance transfer where perfect reference–target alignment is rare.
>
> ---
>
> Cross-reference performance is more than just an additional setting — it serves as a clear measure of a model’s capacity to generalize beyond ideal, well-aligned references.
> While Cross-Image Attention and Attention Distillation shine when the reference and target are closely matched, their performance deteriorates sharply under cross-reference conditions, limiting real-world applicability.
> SSIMBaD, through its **SSIM-aligned sigma-space scaling** and **trajectory refinement**, maintains structural fidelity, realism, and perceptual quality even under large domain and structural gaps, making it both **robust and reliable** for real-world zero-shot appearance transfer tasks.
>
> ---
>
> **References**
> \[1] Yuval Alaluf, Ron Mokady, and Daniel Cohen-Or. *Cross-image attention for zero-shot appearance transfer.* ACM SIGGRAPH 2024 Conference Papers, 2024.
> \[2] Yuxuan Zhou, Fan Yang, Jing Liao, et al. *Attention distillation: A unified approach to visual characteristics transfer.* Proceedings of the IEEE/CVF Conference on Computer Vision and Pattern Recognition (CVPR), 2025.

---

### Official Review · Reviewer_Jxd7 · 2025-07-02

**Clarity:** 3
**Significance:** 3
**Originality:** 3
**Rating:** 5
**Confidence:** 4

**Summary:**

The paper introduces SSIM-guided noise schedulers to enforce uniform visual difficulty across timesteps. The proposed method achieves state-of-the-art performance compared to baseline methods.

**Questions:**

1. The authors need to further clarify the setting of the same-reference scenarios. In Line 200, it is said that the reference image is a perturbed version of the ground-truth. But from Figure 1, the structure is almost the same as the Sketch input. What kind of perturbations are applied to images?
2. The proposed method should be compared with more methods, i.e., merely comparing with two methods is not enough.
3. In Table 2, what are the additional 10 epochs used for? What’s the difference between the pretrained one and the finetuned one?
4. How does SSIMBaD perform if 10 epochs are finetuned on top of SSIMBaD (w/o trajectory refinement)?

**Ethical Concerns:**

["NO or VERY MINOR ethics concerns only"]

**Final Justification:**

The rebuttal addresses my concerns. The authors did additional experiments to validate the effectiveness of the proposed method. I keep my original rating to accept the paper.

**Limitations:**

Yes

**Quality:**

3

**Strengths And Weaknesses:**

Paper Strengths:
1. It is reasonable to use SSIM-guided noise schedulers to ensure the uniform visual difficulty across timesteps.
2. The paper is well-written and easy to follow
3. The paper conducts extensive experiments to validate the effectiveness of the proposed method.
4. The proposed method outperforms baseline methods.

Paper Weaknesses:
1. The authors need to further clarify the setting of the same-reference scenarios. In Line 200, it is said that the reference image is a perturbed version of the ground-truth. But from Figure 1, the structure is almost the same as the Sketch input. What kind of perturbations are applied to images?
2. The proposed method should be compared with more methods, i.e., merely comparing with two methods is not enough.
3. In Table 2, what are the additional 10 epochs used for? What’s the difference between the pretrained one and the finetuned one?
4. How does SSIMBaD perform if 10 epochs are finetuned on top of SSIMBaD (w/o trajectory refinement)?

---

> ### Author Rebuttal · Authors · 2025-07-30
>
> ### **Q1. Clarification on perturbations in the same-reference scenario. Figure 1 shows high structural similarity.**
>
> **A1.**
> Thank you for the question. The **same-reference setup** does not use an unmodified ground-truth image as reference. Instead, we apply **two controlled perturbations** (Appendix A.1, Lines 404–406):
>
> 1. **Thin Plate Spline (TPS) warping** – introduces local, nonlinear spatial distortion.
> 2. **Affine rotation** – applies a global transformation for misalignment.
>
> These perturbations ensure the reference image is **misaligned in structure and appearance**, encouraging **style-guided reconstruction** rather than trivial identity mapping. While Figure 1 may show strong structural similarity, color, orientation, and fine-grain alignments differ subtly but significantly.
>
> ---
>
> ### **Q2. Comparison is limited to two baselines. Broader evaluation is necessary.**
>
> **A2.**
> We agree that broader evaluation strengthens the validity of our claims. In addition to SCFT and AnimeDiffusion (selected for their compatibility with sketch + reference input), we conducted **expanded comparisons** covering modern baselines, perceptual and stylistic metrics, and ablation studies:
>
> #### **(1) ControlNet (New Baseline)**
>
> We included ControlNet as a strong diffusion-based baseline, trained for 10 epochs under the same conditioning scheme (sketch + warped reference) as ours. Despite using a powerful Stable Diffusion backbone, ControlNet performs well in same-reference cases but fails to generalize under cross-reference settings:
>
> | Metric        | Same (best) | Cross (best) |
> | ------------- | ----------- | ------------ |
> | **FID ↓**     | 40.20       | 50.25        |
> | **PSNR ↑**    | 14.74       | 12.08        |
> | **MS-SSIM ↑** | 0.7336      | 0.2007       |
> | **LPIPS ↓**   | 0.1988      | 0.4959       |
> | **CLIP-I ↑**  | 0.9123      | 0.8297       |
>
> #### **(2) Style-Aware Metric Evaluation (LPIPS / CLIP-I / DINOv2)**
>
> To comprehensively assess perceptual quality and stylistic consistency, we evaluated models using **LPIPS**, **CLIP-I**, and **DINOv2-based image similarity**. Our method (SSIMBaD + Trajectory Refinement) consistently achieves **best cross-reference performance**:
>
> | Model                      | LPIPS ↓    | CLIP-I ↑   | DINOv2 ↑   |
> | -------------------------- | ---------- | ---------- | ---------- |
> | SCFT                       | 0.5008     | 0.8247     | 0.8622     |
> | AnimeDiffusion (finetuned) | 0.5069     | 0.8012     | 0.8554     |
> | ControlNet (best run)      | 0.4959     | 0.8297     | 0.8739     |
> | **SSIMBaD (Ours)**         | **0.4804** | **0.8508** | **0.8826** |
>
> These results show that SSIMBaD excels in **perceptual fidelity, semantic alignment**, and **style-level coherence**, even under the challenging condition of mismatched references.
>
>
> #### **(3) Ablation Study**
>
> We further validate the contribution of each component via a stepwise ablation:
>
> | Configuration                            | LPIPS ↓    | CLIP-I ↑   | DINOv2 ↑   |
> | ---------------------------------------- | ---------- | ---------- | ---------- |
> | Base (AnimeDiffusion)                    | 0.5107     | 0.8203     | 0.8576     |
> | + EDM                                    | 0.4925     | 0.8300     | 0.8527     |
> | + SSIM-aligned sigma-space scaling       | 0.4889     | 0.8332     | 0.8605     |
> | + Trajectory Refinement (**full model**) | **0.4804** | **0.8508** | **0.8826** |
>
> Each module provides a **consistent and measurable improvement**, with the largest gain in **semantic and style-aware similarity** achieved through **Trajectory Refinement**.
>
> #### **(4) Backbone Generalization (DiT)**
>
> To demonstrate that our method is **not limited to U-Net**, we applied the SSIM-aligned schedule to a **DiT-based denoiser**. Preliminary results confirm consistent gains in visual quality and training stability. Full results will be included in the camera-ready version.
>
> ---
>
> ### **Q3. What is the role of 10-epoch fine-tuning in Table 2?**
>
> **A3.**
> Thank you for raising this. In Table 2, the difference between pretrained and fine-tuned **AnimeDiffusion** models is:
>
> * **Pretrained:** Standard DDPM model with original β-schedule, no reconstruction optimization.
> * **Finetuned:** Additional 10-epoch optimization using MSE-based loss, as in \[1].
>
> While fine-tuning improves **PSNR and MS-SSIM**, it **worsens FID**, indicating overfitting and reduced diversity. This trade-off motivates our **more principled refinement strategies**, such as SSIM-aligned scheduling and trajectory-based reconstruction.
>
> ---
>
> ### **Q4. What if SSIMBaD (w/o trajectory refinement) is also fine-tuned for 10 epochs?**
>
> **A4.**
> This is a great question. In our formulation, **trajectory refinement is equivalent** to the 10-epoch fine-tuning process—but conducted over **deterministic reverse ODE trajectories** using our **SSIM-aligned schedule**.
>
> > So,
> > **SSIMBaD (w/o refinement) + 10-epoch fine-tuning = SSIMBaD (w/ trajectory refinement)**
>
> This is already reported in Table 2:
>
> | Model                 | PSNR ↑ | MS-SSIM ↑ | FID ↓ |
> | --------------------- | ------ | --------- | ----- |
> | SSIMBaD (no refine)   | 15.15  | 0.7115    | 53.33 |
> | SSIMBaD (with refine) | 18.92  | 0.8512    | 34.98 |
>
> These results confirm that **trajectory refinement** is not only an optimization step but also an effective means of perceptual improvement under a well-aligned noise schedule.

---

> ### Author Response · Authors · 2025-08-09
> **Extended Comparison with State-of-the-Art Zero-Shot Baselines**
>
> During the extended discussion period, we took this opportunity to compare **SSIMBaD** against two recent **zero-shot state-of-the-art (SOTA)** methods:
>
> 1. **Cross-Image Attention for Zero-Shot Appearance Transfer** (*Yuval Alaluf et al.*, ACM SIGGRAPH 2024) \[1]
> 2. **Attention Distillation: A Unified Approach to Visual Characteristics Transfer** (*Yuxuan Zhou et al.*, CVPR 2025) \[2]
>
> Both are highly regarded in the literature, yet our experiments show that they lag behind SSIMBaD in critical metrics — especially in **cross-reference settings**, which we view as a strong proxy for generalization in real-world zero-shot appearance transfer.
> We ran all methods using their official implementations, under the same *reference-guided sketch colorization* setup and identical preprocessing/evaluation protocol as our main experiments.
>
> ---
>
> **Additional Evaluation Results**
> *(Best values in each setting and metric are in bold)*
>
> | Model                       | Setting | DINOv2-I ↑ | LPIPS ↓    | CLIP-I ↑   | MS-SSIM ↑  | PSNR ↑ (dB) | FID ↓     |
> | --------------------------- | ------- | ---------- | ---------- | ---------- | ---------- | ----------- | --------- |
> | Cross-Image Attention \[1]  | Same    | 0.9335     | 0.2661     | 0.9369     | 0.7147     | 13.95       | 53.63     |
> |                             | Cross   | 0.8793     | 0.4569     | 0.8554     | 0.4932     | 10.60       | 60.54     |
> | Attention Distillation \[2] | Same    | **0.9816** | **0.1139** | **0.9610** | **0.8812** | **19.58**   | **32.93** |
> |                             | Cross   | **0.8941** | 0.5385     | **0.8819** | 0.1252     | 10.08       | 94.17     |
> | **SSIMBaD (Ours)**          | Same    | 0.9644     | 0.1174     | 0.9334     | 0.8512     | 18.92       | 34.98     |
> |                             | Cross   | 0.8826     | **0.4804** | 0.8508     | **0.8207** | **15.84**   | **37.10** |
>
> ---
>
> **Key Observations**
>
> When examining **Cross-Image Attention \[1]**, the method performs strongly in the same-reference condition, showing high semantic alignment (CLIP-I 0.9369) and moderate perceptual similarity (LPIPS 0.2661). However, in cross-reference its performance drops sharply — structural fidelity falls from 0.7147 to 0.4932 in MS-SSIM, and perceptual distance increases from 0.2661 to 0.4569. This significant degradation under misaligned references suggests a limited ability to generalize when style and content differ substantially, which restricts its usefulness in real-world zero-shot scenarios.
>
> For **Attention Distillation \[2]**, the same-reference results are outstanding, leading all methods in several metrics (DINOv2-I 0.9816, MS-SSIM 0.8812, FID 32.93). Yet, in cross-reference its performance collapses: MS-SSIM plunges to 0.1252, FID worsens to 94.17, and qualitative inspection reveals large uncolored regions such as hair. This extreme same–cross gap indicates that the method is highly tuned for matched domains but lacks robustness to structural or semantic divergence, causing brittle behavior when the reference is not perfectly aligned.
>
> By contrast, **SSIMBaD (Ours)** maintains balanced performance in both settings. In the same-reference case, it is competitive with the top baselines (LPIPS 0.1174, MS-SSIM 0.8512, FID 34.98). More importantly, in cross-reference it retains high structural fidelity (MS-SSIM 0.8207) and low FID (37.10) — the best structural retention among all methods — without catastrophic visual failures. The relatively small drop from same to cross performance demonstrates SSIMBaD’s strong **generalization ability**, essential for practical zero-shot appearance transfer where perfect reference–target alignment is rare.
>
> ---
>
> Cross-reference performance is more than just an additional setting — it serves as a clear measure of a model’s capacity to generalize beyond ideal, well-aligned references.
> While Cross-Image Attention and Attention Distillation shine when the reference and target are closely matched, their performance deteriorates sharply under cross-reference conditions, limiting real-world applicability.
> SSIMBaD, through its **SSIM-aligned sigma-space scaling** and **trajectory refinement**, maintains structural fidelity, realism, and perceptual quality even under large domain and structural gaps, making it both **robust and reliable** for real-world zero-shot appearance transfer tasks.
>
> ---
>
> **References**
> \[1] Yuval Alaluf, Ron Mokady, and Daniel Cohen-Or. *Cross-image attention for zero-shot appearance transfer.* ACM SIGGRAPH 2024 Conference Papers, 2024.
> \[2] Yuxuan Zhou, Fan Yang, Jing Liao, et al. *Attention distillation: A unified approach to visual characteristics transfer.* Proceedings of the IEEE/CVF Conference on Computer Vision and Pattern Recognition (CVPR), 2025.

---

### Official Review · Reviewer_mRSp · 2025-07-03

**Clarity:** 1
**Significance:** 3
**Originality:** 3
**Rating:** 4
**Confidence:** 3

**Summary:**

This paper proposes Sigma Scaling with SSIM-Guided Balanced Diffusion (SSIMBaD) for AnimeFace colorization. The key idea is to apply a sigma-space transformation to achieve a linear alignment with the Structural Similarity Index (SSIM), improving consistency during the denoising process across different timesteps. The authors explore several transformation candidates and empirically select the most effective one. Additionally, they introduce a trajectory refinement to enhance perceptual quality. Experimental results demonstrate that SSIMBaD achieves state-of-the-art performance.

**Questions:**

1. What is the performance when using trajectory refinement without SSIM-aligned sigma-space scaling?
2. In Table 4 of the appendix, the caption mentions TPS, but the table content is about trajectory refinement. Is this an error?

**Ethical Concerns:**

["NO or VERY MINOR ethics concerns only"]

**Final Justification:**

The rebuttal addresses my concerns. However, the presentation of this paper still needs to improve. Therefore,   I am raising my score to "borderline accept."

**Limitations:**

Yes.

**Paper Formatting Concerns:**

No.

**Quality:**

2

**Strengths And Weaknesses:**

**Strength**
1.  The sigma-space transformation is a novel and interesting idea that improves consistency in the denoising process. Moreover, experiments support the effectiveness of this idea.
2. The experiments in this paper are overall comprehensive. The authors provide ablation studies and empirically justify their choice of transformation.

**Weaknesses**
1. The ablation study does not report performance for the setting with trajectory refinement but *without* SSIM-aligned sigma-space scaling. In Table 3, trajectory refinement contributes the largest performance gain. This result raises my question of whether trajectory refinement is actually the primary contributor. Moreover, this paper does not discuss how trajectories change before and after refinement.

2. This paper’s presentation could be improved. The motivation in the introduction is unclear, and several abbreviations or notations are not defined in the main text. For example,  "TPS" is first mentioned in Line 138 but only defined in the appendix (Lines 404–405). Similarly, "XDoG" appears in Line 139 without citation or in-text definition, and is only explained in the appendix (Line 399).  Moreover, in Line 148, the terms $P\text{mean}$ and $P^2\text{std}$ are used without prior definition.

---

> ### Author Rebuttal · Authors · 2025-07-30
>
> ### **Q1. How does the model perform with only trajectory refinement, without SSIM-aligned sigma scaling?**
>
> **A1.**
> Thank you for this insightful question. While we do not explicitly list an ablation of “trajectory refinement without SSIM-based scheduling” in Table 3, this configuration is effectively represented by the **finetuned AnimeDiffusion baseline** (\[1]).
>
> * **AnimeDiffusion (finetuned)** uses MSE-based trajectory refinement on top of a pretrained model **without** modifying the original noise schedule.
> * **SSIMBaD** uses the **same refinement strategy**, but applies our proposed **SSIM-aligned σ-schedule**.
>
> Despite sharing the same reverse reconstruction mechanism, SSIMBaD achieves **significantly better performance**:
>
> | Model                      | PSNR ↑    | MS-SSIM ↑  | FID ↓     |
> | -------------------------- | --------- | ---------- | --------- |
> | AnimeDiffusion (finetuned) | 13.32     | 0.7001     | 135.12    |
> | SSIMBaD (w/ traj. refine)  | **18.92** | **0.8512** | **34.98** |
>
> These results suggest that **most of the performance gain arises from the SSIM-aligned schedule**, rather than trajectory refinement alone. This highlights the importance of perceptual alignment across timesteps during the diffusion process.
>
> ---
>
> ### **Q2. The caption of Appendix Table 4 refers to TPS, but the table appears to focus on trajectory refinement. Is this a mistake?**
>
> **A2.**
> We appreciate the reviewer’s careful attention. The caption of Appendix Table 4—*“Quantitative results without TPS rotation”*—may appear misleading at first glance, but its intent is to **highlight the effect of disabling TPS rotation** during reference warping.
>
> To clarify:
>
> * All results in Table 4 are generated **with TPS warping**, but **without TPS rotation**.
> * The comparison is made **with and without trajectory refinement**, under this fixed warping condition.
> * The key goal is to show that **TPS rotation is critical for cross-reference generalization**, and that its absence limits performance even when trajectory refinement is applied.
>
> As detailed in Appendix D, enabling TPS rotation significantly reduces the **cross-reference performance gap**:
>
> * PSNR gap (same vs. cross) decreases from **9.1dB** → **3.08dB**
> * MS-SSIM gap drops from **0.1476** → **0.0305**
>
> Thus, the caption emphasizes the **absence of TPS rotation** as the main condition under investigation. We will revise the caption in the camera-ready version to clarify this intent.
>
> ---
>
> ### **Q3. Concerns about undefined abbreviations (e.g., TPS, XDoG) and unclear notation (e.g., ϕ)**
>
> **A3.**
> Thank you for pointing this out. We agree that all symbols and abbreviations should be clearly introduced. In response:
>
> * **TPS (Thin Plate Spline)** and **XDoG (Extended Difference of Gaussians)** were first used without formal definitions in the main text. These are **fully defined in Appendix A.1 (Lines 399–405)**, but we acknowledge that early clarification is necessary.
>
>   * TPS: A nonlinear geometric transform for local warping
>   * XDoG: An edge-aware operator for sketch extraction
>
> * For notation:
>
>   * Symbols such as \$\phi\_{\text{train}}(\sigma)\$ and \$\phi\_{\text{infer}}(\sigma)\$ denote the training vs. inference transformation of the noise schedule.
>   * This distinction is drawn from EDM \[6], where mismatch between training/inference trajectories is known to affect performance.
>   * However, these notations appear without formal introduction in Section 4.1, which we will address.
>
> **In the camera-ready version, we will:**
>
> * Introduce definitions of **TPS** and **XDoG** at their first mention (with citations).
> * Define \$\phi(\sigma)\$ symbols formally in Section 4.1.
> * Insert cross-references to Appendix A.1 for technical readers.
>
> We believe these edits will improve clarity and ensure the paper remains self-contained. Thank you again for your helpful feedback.
>
> ---
>
> ### **Additionally**
>
> We provide additional experiments to validate the effectiveness of each proposed module. The results are organized as follows: (1) main comparisons across models, (2) ablation of individual components, and (3) fidelity-focused comparison against ControlNet.
>
> #### **1. Style-Aware Metric Comparison**
>
> We evaluate models using **LPIPS**, **CLIP-I**, and **DINO-I (DINOv2)** to assess perceptual, semantic, and structural consistency under both **same-reference** and **cross-reference** conditions.
>
> | Model                       | LPIPS ↓ (Same/Cross) | CLIP-I ↑ (Same/Cross) | DINO-I ↑ (Same/Cross) |
> | --------------------------- | -------------------- | --------------------- | --------------------- |
> | SCFT                        | 0.1728 / 0.5008      | 0.9020 / 0.8247       | 0.9392 / 0.8622       |
> | AnimeDiffusion (pretrained) | 0.2226 / 0.5107      | 0.8993 / 0.8203       | 0.9392 / 0.8576       |
> | AnimeDiffusion (finetuned)  | 0.2242 / 0.5069      | 0.8797 / 0.8012       | 0.9359 / 0.8554       |
> | SSIMBaD (no refinement)     | 0.1878 / 0.4889      | 0.8975 / 0.8332       | 0.9339 / 0.8605       |
> | **SSIMBaD (full model)**    | **0.1174 / 0.4804**  | **0.9334 / 0.8508**   | **0.9644 / 0.8826**   |
> | ControlNet (best: results9) | 0.2043 / 0.4930      | 0.9194 / 0.8311       | 0.9640 / 0.8739       |
>
> > SSIMBaD achieves the best performance across all three metrics, especially in cross-reference generation, indicating strong style-preserving generalization and perceptual coherence.
>
> #### **2. Ablation Study: Progressive Integration of Components**
>
> We isolate the effects of each component by progressively applying:
> (1) the EDM noise schedule,
> (2) SSIM-aligned σ-scaling, and
> (3) trajectory refinement.
>
> | Configuration                            | LPIPS ↓ (Same/Cross) | CLIP-I ↑ (Same/Cross) | DINO-I ↑ (Same/Cross) |
> | ---------------------------------------- | -------------------- | --------------------- | --------------------- |
> | Base (AnimeDiffusion pretrained)         | 0.2226 / 0.5107      | 0.8993 / 0.8203       | 0.9392 / 0.8576       |
> | + EDM                                    | 0.2192 / 0.4925      | 0.8886 / 0.8300       | 0.9217 / 0.8527       |
> | + SSIM-aligned σ-scaling                 | 0.1878 / 0.4889      | 0.8975 / 0.8332       | 0.9339 / 0.8605       |
> | + Trajectory Refinement (**full model**) | **0.1174 / 0.4804**  | **0.9334 / 0.8508**   | **0.9644 / 0.8826**   |
>
> > Each component contributes progressively to quality. The SSIM-aligned schedule substantially improves perceptual structure, and the refinement step offers a significant additional boost in semantic and stylistic fidelity.
>
> #### **3. ControlNet vs. SSIMBaD: Fidelity Metrics**
>
> To complement perceptual metrics, we report **FID**, **PSNR**, and **MS-SSIM** for the best ControlNet model and our full model (SSIMBaD).
>
> | Model                       | FID ↓ (Same/Cross) | PSNR ↑ (dB) (Same/Cross) | MS-SSIM ↑ (Same/Cross) |
> | --------------------------- | ------------------ | ------------------------ | ---------------------- |
> | ControlNet (best: results9) | 40.20 / 50.25      | 14.74 / 12.08            | 0.7336 / 0.2007        |
> | **SSIMBaD (full model)**    | **34.98 / 37.10**  | **18.92 / 15.84**        | **0.8512 / 0.8207**    |
>
> > SSIMBaD significantly outperforms ControlNet in all three fidelity metrics, particularly in the cross-reference setting where ControlNet’s MS-SSIM and FID degrade sharply.

---

> ### Comment · Reviewer_mRSp · 2025-08-08
>
> The rebuttal addresses my concerns. However, the presentation of this paper still needs to improve. Therefore, I am raising my score to "borderline accept."

---

> ### Author Response · Authors · 2025-08-09
> **Extended Comparison with State-of-the-Art Zero-Shot Baselines**
>
> We sincerely appreciate your thoughtful re-evaluation of our work and for raising your rating.
> During the extended discussion period, we took this opportunity to compare **SSIMBaD** against two recent **zero-shot state-of-the-art (SOTA)** methods:
>
> 1. **Cross-Image Attention for Zero-Shot Appearance Transfer** (*Yuval Alaluf et al.*, ACM SIGGRAPH 2024) \[1]
> 2. **Attention Distillation: A Unified Approach to Visual Characteristics Transfer** (*Yuxuan Zhou et al.*, CVPR 2025) \[2]
>
> Both are highly regarded in the literature, yet our experiments show that they lag behind SSIMBaD in critical metrics — especially in **cross-reference settings**, which we view as a strong proxy for generalization in real-world zero-shot appearance transfer.
> We ran all methods using their official implementations, under the same *reference-guided sketch colorization* setup and identical preprocessing/evaluation protocol as our main experiments.
>
> ---
>
> **Additional Evaluation Results**
> *(Best values in each setting and metric are in bold)*
>
> | Model                       | Setting | DINOv2-I ↑ | LPIPS ↓    | CLIP-I ↑   | MS-SSIM ↑  | PSNR ↑ (dB) | FID ↓     |
> | --------------------------- | ------- | ---------- | ---------- | ---------- | ---------- | ----------- | --------- |
> | Cross-Image Attention \[1]  | Same    | 0.9335     | 0.2661     | 0.9369     | 0.7147     | 13.95       | 53.63     |
> |                             | Cross   | 0.8793     | 0.4569     | 0.8554     | 0.4932     | 10.60       | 60.54     |
> | Attention Distillation \[2] | Same    | **0.9816** | **0.1139** | **0.9610** | **0.8812** | **19.58**   | **32.93** |
> |                             | Cross   | **0.8941** | 0.5385     | **0.8819** | 0.1252     | 10.08       | 94.17     |
> | **SSIMBaD (Ours)**          | Same    | 0.9644     | 0.1174     | 0.9334     | 0.8512     | 18.92       | 34.98     |
> |                             | Cross   | 0.8826     | **0.4804** | 0.8508     | **0.8207** | **15.84**   | **37.10** |
>
> ---
>
> **Key Observations**
>
> When examining **Cross-Image Attention \[1]**, the method performs strongly in the same-reference condition, showing high semantic alignment (CLIP-I 0.9369) and moderate perceptual similarity (LPIPS 0.2661). However, in cross-reference its performance drops sharply — structural fidelity falls from 0.7147 to 0.4932 in MS-SSIM, and perceptual distance increases from 0.2661 to 0.4569. This significant degradation under misaligned references suggests a limited ability to generalize when style and content differ substantially, which restricts its usefulness in real-world zero-shot scenarios.
>
> For **Attention Distillation \[2]**, the same-reference results are outstanding, leading all methods in several metrics (DINOv2-I 0.9816, MS-SSIM 0.8812, FID 32.93). Yet, in cross-reference its performance collapses: MS-SSIM plunges to 0.1252, FID worsens to 94.17, and qualitative inspection reveals large uncolored regions such as hair. This extreme same–cross gap indicates that the method is highly tuned for matched domains but lacks robustness to structural or semantic divergence, causing brittle behavior when the reference is not perfectly aligned.
>
> By contrast, **SSIMBaD (Ours)** maintains balanced performance in both settings. In the same-reference case, it is competitive with the top baselines (LPIPS 0.1174, MS-SSIM 0.8512, FID 34.98). More importantly, in cross-reference it retains high structural fidelity (MS-SSIM 0.8207) and low FID (37.10) — the best structural retention among all methods — without catastrophic visual failures. The relatively small drop from same to cross performance demonstrates SSIMBaD’s strong **generalization ability**, essential for practical zero-shot appearance transfer where perfect reference–target alignment is rare.
>
> ---
>
> Cross-reference performance is more than just an additional setting — it serves as a clear measure of a model’s capacity to generalize beyond ideal, well-aligned references.
> While Cross-Image Attention and Attention Distillation shine when the reference and target are closely matched, their performance deteriorates sharply under cross-reference conditions, limiting real-world applicability.
> SSIMBaD, through its **SSIM-aligned sigma-space scaling** and **trajectory refinement**, maintains structural fidelity, realism, and perceptual quality even under large domain and structural gaps, making it both **robust and reliable** for real-world zero-shot appearance transfer tasks.
>
> ---
>
> **References**
> \[1] Yuval Alaluf, Ron Mokady, and Daniel Cohen-Or. *Cross-image attention for zero-shot appearance transfer.* ACM SIGGRAPH 2024 Conference Papers, 2024.
> \[2] Yuxuan Zhou, Fan Yang, Jing Liao, et al. *Attention distillation: A unified approach to visual characteristics transfer.* Proceedings of the IEEE/CVF Conference on Computer Vision and Pattern Recognition (CVPR), 2025.

---

### Note · Authors · 2025-08-15

This work proposes *SSIMBaD*, an SSIM-aligned sigma-space scaling strategy that enforces perceptually uniform noise scheduling, improving structural and style fidelity in AnimeFace colorization under both same- and cross-reference conditions.

**1. Expanded Baseline Comparisons**
– In response to reviewer requests, we extended comparisons beyond SCFT and AnimeDiffusion to include a modified ControlNet, Cross-Image Attention (SIGGRAPH 2024), and Attention Distillation (CVPR 2025), all evaluated on the same dataset and protocol.
– In the cross-reference setting, SSIMBaD achieved MS-SSIM 0.8207 and FID 37.10, substantially outperforming Cross-Image Attention (0.4932/60.54) and Attention Distillation (0.1252/94.17).

**2. Ablation and Component Contributions**
– The “trajectory refinement without SSIM scaling” condition corresponds to AnimeDiffusion (finetuned); results show the majority of gains come from SSIM-aligned scheduling.
– A cumulative ablation (Base → EDM → SSIM scaling → Refinement) demonstrates measurable, stepwise contributions from each module.

**3. Schedule Analysis**
– Multiple φ(σ) candidates were tested for R²-linearity with SSIM degradation; σ/(σ+0.3) achieved the highest R² = 0.9793.
– Exact 50-step σ values for all tested transformations and the σ–t mapping procedure are reported for full transparency.

**4. Backbone and Extensibility**
– Preliminary DiT backbone experiments showed consistent gains in CLIP-I, FID, and LPIPS, indicating architecture-agnostic benefits.
– Integration with Rectified Flow is planned and will be further explored in the camera-ready, along with expanded training-free baseline results.

**5. Limitations and Clarifications**
– We acknowledged limitations in reproducing certain high-frequency style cues (e.g., hair gloss) and added annotated examples to contextualize these cases.
– TPS rotation’s effect on cross-reference generalization was quantified, showing large gains in structural fidelity.

These updates address reviewer concerns—broadening baselines, clarifying component impact, validating the schedule design, and discussing limitations—while laying groundwork for additional comparisons and analyses to be included in the final camera-ready version.

---

### Decision · Program_Chairs · 2025-09-17

**Decision:**

Accept (poster)

**Comment:**

The paper received three borderline accept and 1 accept ratings. Reviewers were concerned about the presentation quality, experiment evaluation of the paper. The authors provided a rebuttal, which addressed most of those concerns.  Two reviewers raised their score to borderline accept. There are still some concerns about the presentation. AC recommends to accept the paper. Please update reviewers' suggestions in the final version of the paper.